# Structural mechanism of bacteriophage lambda tail's interaction with the bacterial receptor

Xiaofei Ge [1] & Jiawei Wang [1] ✉

Bacteriophage infection, a pivotal process in microbiology, initiates with the phage's tail recognizing and binding to the bacterial cell surface, which then mediates the injection of viral DNA. Although comprehensive studies on the interaction between bacteriophage lambda and its outer membrane receptor, LamB, have provided rich information about the system's biochemical properties, the precise molecular mechanism remains undetermined. This study revealed the high-resolution cryo-electron microscopy (cryo-EM) structures of the bacteriophage lambda tail complexed with its irreversible *Shigella sonnei 3070* LamB receptor and the closed central tail fiber. These structures reveal the complex processes that trigger infection and demonstrate a substantial conformational change in the phage lambda tail tip upon LamB binding. Providing detailed structures of bacteriophage lambda infection initiation, this study contributes to the expanding knowledge of lambda-bacterial interaction, which holds significance in the fields of microbiology and therapeutic development.

Bacteriophage lambda, a member of the *Siphovir*uses with a long and flexible tail, has served as a crucial model for bacteriophage biology and genetics since its discovery in the 1950s[1–3]. The lambda phage lifecycle initiates with host recognition, proceeds with DNA injection, and determines either replication or lysogeny. This process culminates in the host cell lysis, releasing new phage particles[4]. A key step of this lifecycle is the initial interaction between the bacteriophage and its host, typically a multistep recognition process[5–7]. Although the original bacteriophage lambda isolates initially attach to its host with the assistance of side fibers, which is referred to as its primary receptor interaction, common laboratory strains like λWT, absence of the side fibers due to a mutation in the side tail fiber (stf) gene, were proved to maintain the basic ability to absorb and infect hosts[8,9]. This indicates that the downstream receptor interaction is the decisive factor of lambda phage absorption and infection, which is assumed by lambda central tail fiber and host's outer membrane protein, LamB[9], which is also utilized by other phages such as K10[10] and TP1[11], for irreversible adsorption[12,13].

Bacteriophage lambda's tip protein, gpJ, plays a crucial role when attaching to the cell surface[14]. The gpJ protein, comprised of 1132 residues, is located at the bottom end of the lambda phage tail, serving as a key component of the lambda baseplate[15]. The gpJ trimer not only seals the tail tip into a closed cone but also extends a central tail fiber, providing the means through which lambda phage directly interacts with its host. Genetic evidence has identified the *E. coli* receptor LamB (ecLamB) as the direct target interacting with gpJ during the phage's attachment process[16]. When the J gene from phage lambda (gpJ) was replaced with the tail fiber gene from bacteriophage 434, the modified phages can bind to OmpC, the membrane receptor that phage 434 utilizes for infection[14]. Further studies indicate that the C-terminal of gpJ protein determines the host specificity of the phage[17]. Electron microscopy imaging of bacteriophage lambda incubated with *E. coli* LamB protein incorporated into liposomes revealed two distinct types of bacteriophage lambda-ecLamB complexes[7]. In one type, only the end of the phage tail (gpJ) is bound to the liposome. In the other, the bottom end of the tail tube is irreversibly attached to the liposome. When ecLamB is replaced with the *Shigella sonnei 3070* LamB variant (ssLamB)—which only differs from ecLamB in seven amino acids within the 381–390 region—the resultant structural change triggers

[1]State Key Laboratory of Membrane Biology, Beijing Frontier Research Center for Biological Structure, School of Life Sciences, Tsinghua University, 100084 Beijing, PR China. ✉e-mail: jwwang@tsinghua.edu.cn

spontaneous in vitro DNA ejection from the bacteriophage, an irreversible event[16–18].

Following the initial interaction of phage lambda and LamB, a transmembrane channel needs to form to traverse the two membranes and periplasmic space, facilitating efficient DNA passage[18]. The diameter of double-stranded B-DNA is about 20 Å. In contrast, each subunit of trimeric LamBs consists of an 18-stranded, pore-forming β-barrel, which forms a water-filled channel with a diameter of 5–6 Å. LamB is selectively permeable, primarily favoring the passage of maltose and maltodextrins, while also permitting other small hydrophilic molecules to a lesser extent. Given the narrow diameter and rigid structure of the LamB pore, it is unlikely that the double-stranded DNA can pass through it[19,20]. It is plausible that phage proteins, rather than LamB itself, facilitate the formation of the DNA channel across the outer membrane. As the phage protein interacts with LamB, gpJ performs a pivotal role in the state transition, triggering the change from the closed to the open state upon binding. The specific transformations that gpJ undergoes during these two states, especially how it manages to open its part of the channel in the open state as opposed to maintaining a closed configuration, remain areas for further exploration.

For bacteriophage lambda, the high-resolution structure for the closed state of the tail tube is available, but the high-resolution structure of the bottom end of the central tail fiber, as well as the open state structure, is missing[15]. To understand how the lambda phage's receptor facilitates its opening and to gain insights into the structure of the open state, the high-resolution structure of the lambda phage's central tail fiber in its closed state and the structure of the opened lambda phage tail associated with ssLamB were determined in this study (Fig. 1a).

## Results

### Cryo-EM reconstruction of the bacteriophage lambda tail with ssLamB in nanodisc

ssLamB was expressed recombinantly in *E. coli* and isolated from the outer membrane. It was then successfully reconstituted into a nanodisc using the membrane scaffold protein (MSP2N2). The purified phage lambda tail was subsequently incubated with the ssLamB nanodisc, leading to the formation of the ssLamB-tail complex. Following extensive cryo-EM image processing, both the tail tube and the density of the FNIIIs of gpJ were successfully obtained. These enabled the visualization

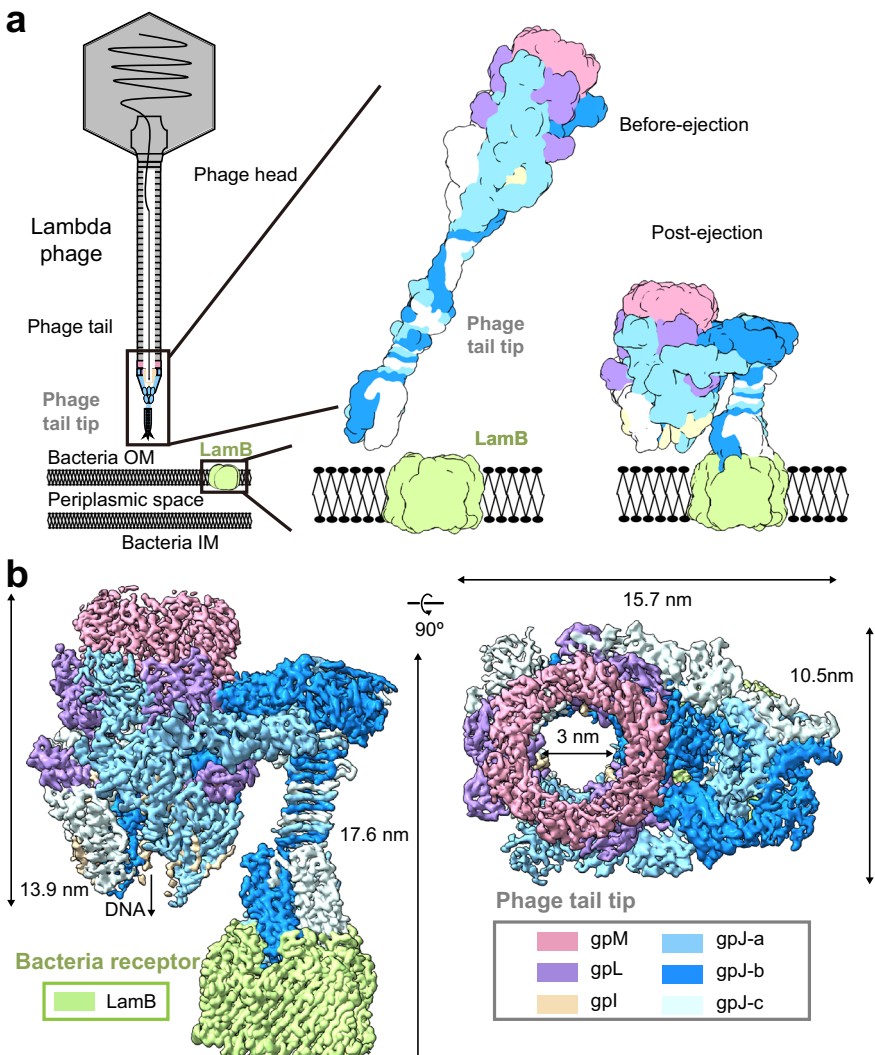

**Fig. 1 | Global view of bacteriophage lambda tail after interaction with LamB in nanodisc. a** Schematic representation of lambda tail interaction with LamB in close/open state. **b** Cryo-EM structure of bacteriophage lambda tail interacting with LamB: sideview of the composite map of the bacteriophage lambda tail in open conformation after interaction with LamB (left), along with a top-down view (right).

This composite map is generated by merging the open state tube map, a locally masked gpJ map, and the gpJ713 with LamB complex map (for visualization purposes only). The figure illustrates a complex composed of the bacteriophage tail and the receptor LamB, including six units of gpM, three units of gpL, three units of gpJ, three units of gpI, and a trimeric LamB.

of the tail tip opening and the deformation of the central tail fiber (Fig. 1b). After further classification of the particles ultimately used, we also obtained a low-resolution map depicting the interaction between the tail tip and LamB within a nanodisc. However, the majority of particles did not fall into this category. This may be attributed to the nanodisc scaffold MSP2N2 being insufficient to simultaneously encapsulate the LamB trimer and the membrane-inserted tube end. While the densities corresponding to the receptor binding domain (RBD) of gpJ and ssLamB were discernible, the resolution was not sufficient for model construction (Supplementary Fig. S1a, b). Therefore, a high-resolution structure of the complex formed between the C-terminal truncation of gpJ, starting from residue 713 (referred to as gpJ713), and ssLamB in a detergent environment was determined (Supplementary Fig. S2a, b). The structure elucidated the interaction between LamB and the lambda tail in further detail. Due to the insufficient resolution of the previously studied bottom part of gpJ, model building was not feasible, making it impossible to analyze and compare with the open structure. A cryo-EM sample of gpJ713 in its closed state alone was also prepared and collected. A 2.76 Å map of the central tail fiber in a closed state was obtained (Supplementary Fig. S2c, d).

Collectively, the four maps reveal the distinct transformations in the lambda phage tail before and after infection (Fig. 1a). Upon LamB binding, the bottom part of gpJ exhibits bending. The tube component demonstrates a descent, coupled with an overall shortening of the entire structure.

### Interaction and selectivity between RBD and LamB

In the lambda phage, the baseplate hub protein, gpJ, comprises a sequence of hub domains (HDs, include HDII-ins1, HDII-ins2, HDII, HDIII, HDIV, OB), two successive FNIIIs (denoted as FNIII-1 and FNIII-2), an α-helical shaft (AHS), a β-sheet prism (central straight fiber, CSF), and a receptor-binding domain (RBD) (Supplementary Fig. S3). The RBD, the C-terminus of gpJ, incorporates an immunoglobulin (Ig)-like domain, with four bottom loops (L1-L4) serving a pivotal role in receptor binding (Fig. 2a). LamB is comprised of three identical subunits, presenting a form similar to a half-open tulip[21]. Each subunit consists of an 18-stranded antiparallel β-barrel (Supplementary Fig. S4a), with three loops (EL1, EL2, and EL3) interacting with the loop EL5 from an adjacent subunit and lined along the inner wall of the barrel to form the channel for small molecules (Supplementary Fig. S4b). The other loops, namely EL4, EL6, and EL9, create a compact structure on the cell surface crucial for the binding of phage lambda. Upon binding with LamB, the distance between the three protomers of RBDs on the bottom side increases from 1.8 nm to 3 nm (Fig. 2b). Correspondingly, the top of RBD narrows, with the top loops moving closer to the central axis (Fig. 2c). The β-sheet above the RBD moves inward, in unison with the loops. Nine new hydrogen bonds form between the second and third layers from the bottom of the β-sheet, reducing the interlayer distance and the distance to the RBD (Fig. 2b). RBD inserts four loops (L1-L4) into the extracellular lumen of LamB (Fig. 2d, e). The interaction surface is significant, with an area of 6195 Å$^2$ buried, a PISA calculated energy of −4.9 kcal/mol, and further stabilized by 45 hydrogen bond interactions. In *S. sonnei*, D255 on loop EL6 contributes a hydrogen bond that stabilizes the conformation of loop EL9 through S384 (Fig. 2f). Biolayer interferometry (BLI) assay shows that *E. coli*'s LamB reversibly binds to gpJ713 with a $K_D$ of 1.76 nM (Fig. 2g). Despite the high structural similarity of their EL9, a single-point mutation (A384S) in ecLamB transitions this interaction to irreversible binding, akin to that of the wild-type LamB from *S. sonnei* (Fig. 2g, Supplementary Fig. S4c).

### Structural alteration in central tail fiber

The central tail fiber of the lambda phage in the C-terminal region of the gpJ protein is constituted by FNIIIs, AHS, CSF, and RBD (Fig. 3a, left). AHS is a helix bundle formed by three α-helices, situated below the FNIIIs, securely anchoring the three FNIII-2 units together from the bottom. The diameter of the tail tube gradually narrows in the region corresponding to the FNIII, eventually closing completely. In the closed state, the longitudinally oriented CSF, a mixed β-sheet prism, exhibits intricate torsion in the strands spanning from above the RBD to below the AHS. From bottom side to top side, its cross-section progressively expands from narrow to wide (Fig. 3a, left). Upon interaction with LamB, the RBD, CSF, and AHS undergo a range of structural rearrangements, from subtle to substantial (Fig. 3a, right, 3b).

Given the described modifications in the RBD and the bottom of the CSF, the inter-domain distance diminishes as a consequence of hydrogen bond formation (Fig. 2b). The CSF undertakes a twist, and its upper region undergoes substantial reorientation and enhanced rotational movement, thereby reducing the twist angle relative to its lower region (Fig. 3c). The AHS situated above the CSF experiences a dramatic orientation shift of approximately 180° (Fig. 3a, right) and enfolds back onto the CSF. The upper α-helix (residues 809–840) within the AHS remodels into a shorter α-helix segment, flanked by an extended two-layered β-sheet at the summit of the CSF (Fig. 3b). The C-terminus of helixes (841–861) in the AHS domains have transformed into two additional β-sheets at the top of the CSF (colored in deep blue, Fig. 3a). The loosen of AHS helix bundle leads to a reduction in the restraints on the FNIII trimer from the bottom side. The linker between the HDs and FNIIIs disengages and repositions, leading to the alignment of two FNIIIs (gpJ-a and gpJ-b) on one side of the CSF, while the third FNIII (gpJ-c) relocates to the opposite side (Fig. 3a, right top). This reconfiguration of the interplay between the FNIII and the tube in gpJ is facilitated by an elongated linker that bridges the HDs of gpJ to the FNIII-1 domain[15]. Following this reorganization, the linker establishes a perpendicular alignment to the tail axis along the HDs of the tail tube.

Compare to the closed state structure, the distance from the FNIIIs to the remote end of the RBD markedly shrinks from 29.7 nm to 13.2 nm. This comprehensive portrayal, representing both open and closed states of the structure, provides an all-encompassing view of the structural transitions in lambda phage from the RBD to the HDs upon LamB binding (Supplementary Movie 1).

### Tail tip baseplate hub domains (BHD) opening mechanism

In siphophages, the tail tip complex (TTC), situated at the bottom end of the tail tube, typically comprises four "Hub Domains" (HDs)[22–25]. For phage lambda, the baseplate hub proteins are represented by the HDs of gpJ and gpL (Fig. 4a). These proteins assemble as a trimer beneath the tail protein gpM ring (Supplementary Fig. S5a, b). The core component of the HDs in gpJ is the HDIV domain, which showcases an adorned OB domain within its loop β5-β6.1. The HDIV's N-arm extends towards the HDII domain, further connecting to the HDII-ins1 in a tandem orientation. HDII-ins2 domain is inserted within the β6-β7 loop of the HDII-ins1 domain, while the HDIV domain substitutes the conventional β3-β4 loop with the HDIII domain. The gpL protein consists of HDI and 4Fe-4S domains (Fig. 4a).

When comparing the TTC structures of phage lambda in closed and open states, significant changes are observed in the structure of the HDs of gpJ. In the closed state, the pseudo-hexametric HDIV-HDI of TTC operates as a bridge between the upper gpM hexamer and the trimeric HDIII-HDII, ensuring the continuity of the tail tube with a diameter of 3 nm (Supplementary Figs. S3b and S5c). Starting from the HDIII-HDII, the diameter progressively narrows to 2 nm until it is ultimately sealed through the joint action of FNIIIs (Supplementary Fig. S5c, white region). Upon the binding of gpJ with LamB, the helix bundle of AHS undergoes a transition, forming a wrap around the upper surface of CSF. This transformation leads to a reduction in the restraints on the FNIII trimer from the bottom side. Subsequently, the FNIIIs dissociate and reposition themselves perpendicular to the

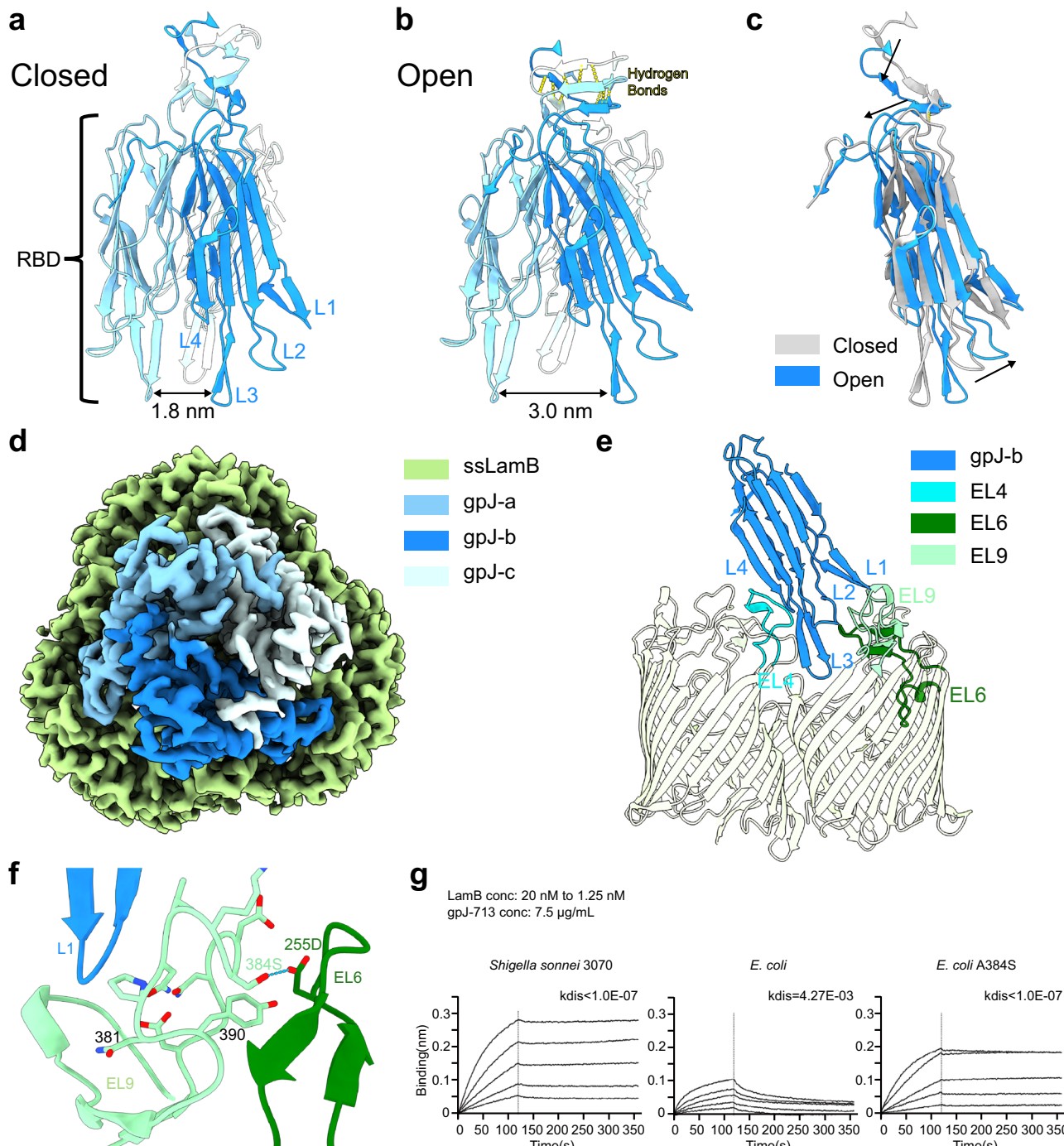

**Fig. 2 | Structural analysis of the RBD-LamB interaction. a** Sideview of the closed state of gpJ RBD. The gpJ RBD is a trimer, with each RBD domain comprising four loops, named L1, L2, L3, and L4. Above the RBD are β-sheets, with a distinctive gap observed between layers. **b** Sideview of the open state of gpJ RBD. An additional nine pairs of hydrogen bonds, formed between the second and third β-strands (counting from the bottom), shorten their distance. **c** Comparison of the closed and open states of the RBD structure. The L1-L4 loops of the RBD shift outward, and the upper part moves closer, with the top β-sheets also approaching the RBD. **d** Topview segment of the map showcasing the receptor binding domain (RBD) of gpJ and LamB within the gpJ713-LamB complex. **e** Highlighting the interacting loops originating from one RBD (L1, L2, L3, L4) and a LamB monomer (EL4, EL6, EL9). **f** Loop EL9 of ssLamB is stabilized through a hydrogen bond with loop EL6 (384S and 255D). **g** Biolayer Interferometry (BLI) of gpJ713 and LamB from *Shigella sonnei* 3070, *E. coli* and *E. coli* S384A mutant. gpJ713 is loaded on the sensor, and five different concentrations of LamB are added. The binding between *Shigella sonnei* 3070's LamB and gpJ713 appears irreversible, whereas its binding with *E. coli* LamB is reversible. The A384S mutation in *E. coli* LamB results in an irreversible binding with gpJ713. Source data are provided as a Source Data file.

HDs, inducing the transition of TTC to an open state. HDII-HDIII, the insertion domains HDII-ins1 and 4Fe-4S rotate away from the tail tip axis (Fig. 4b). Alignment of these domains with their counterparts in the closed state reveals unchanged structures, indicating a rigid rotation as a whole (Supplementary Fig. S5d). Exceptionally,

structure comparison shows that the first stand β1 of the HDII-ins1 domain concurrently swings downward towards the membrane. The HDII-ins2 domain undergoes rotation and determines the narrowest diameter within the HDs region in the final DNA channel (Fig. 4c, Supplementary Fig. S5c).

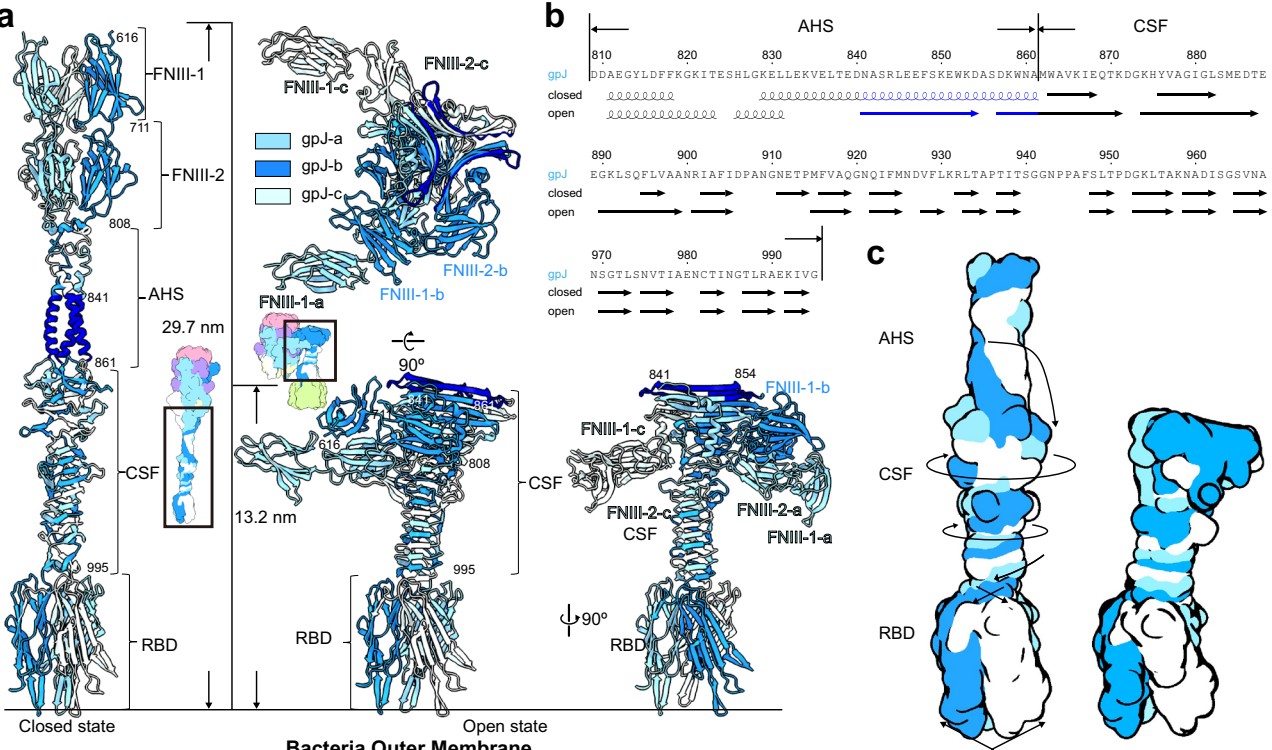

**Fig. 3 | Lambda central tail fiber (gpJ) in both the closed and open states. a** Left: central tail fiber (gpJ) in a closed state, composed of two consecutive FNIIIs (FNIII-1 and FNIII-2), AHS and CSF. Right bottom: central tail fiber different views in an open state. Right top: Top view of FNIII, AHS, and CSF in an open state. Three FNIIIs strings dissociate and reposition, two chains align on one side of the CSF, while the third string migrates to the opposite side. **b** Comparison of the secondary structure of AHS and CSF in the closed and open states. 841-861 colored in deep blue to show which helixes in the AHS domains transformed into additional β-sheets in open state structure. **c** Schematic representation of structural changes in RBD, CSF, and AHS. The binding of RBD to LamB triggers an opening at the bottom, which in turn drives a centripetal motion at the top of RBD. The formation of new hydrogen bonds at the base of CSF results in a more compact CSF structure and induces rotation. The rotation angle is amplified as it propagates upwards, leading to a larger twist at the top of CSF. AHS drops off and wraps around the top of CSF.

## Conformational changes of gpI and tail tube open

In the closed state of the tube, the diameter progressively narrows to 20 Å from the HDIII-HDII region until it is eventually sealed through the combined action of FNIIIs. Inside the baseplate cone, gpI is intercalated within the TTC, blocking the exit for the gpH (tape measure protein) located above gpI, thereby preventing the descent of DNA. The gpI protein consists of several domains: an unresolved domain on the N-terminal, two helices (H1 and H2), a 'plug' domain (159–175) within the tube lumen, and a series of β-strands interconnected by loops (Fig. 5a, b). In the closed state, gpI residues from position 101 onward are resolved, but in the open structure, only residues from position 167 onward can be seen (Fig. 5b). The missing regions, corresponding to the H1 and H2 domains, are hypothesized to have transmembrane capabilities as predicted by TMHMM[26].

In the close state, the plug domain of gpI extends into the tube lumen, forming a blockage, with a loop and β-strands navigating through gaps in gpJ's HDs (Fig. 5a, c). In the open state, the loop and β-strands traversing HD gaps, along with the HDs, exhibit significant positional shifts. This collective outward movement is characterized by uniform direction and distance (Supplementary Fig. S5e). The gpI segment above the gaps descends, inducing a loosening and dropping of the plugs. The residues following the H1 and H2 domains orient towards the outer membrane at the bottom of the tail tube (Fig. 5b). The N-terminal domain remains unseen beneath the TTC. In conclusion, the upward movement of the FNIII, the outward displacement of the HDs, and the outward movement and loosening of the plug domains of gpI expand the diameter of the tail tube to over 3 nm, a size adequate for DNA passage (Fig. 5c, d, e, Supplementary Fig. S5c).

## Discussion

Bacteriophage lambda has been widely utilized as a model system for studying host recognition and infection trigger mechanisms. Using the newly obtained open structure of the phage lambda tail with LamB, the closed structure of the central tail fiber, and the previously published closed structure[15], we suggest a mechanism for lambda phage DNA ejection induced by receptor binding.

In the closed state of phage lambda, HDI and HDIV act as extenders and adaptors of the tail tube, while HDII and HDIII form the bottom portion of the tail, enveloped by FNIIIs. The trimeric plug domains of gpI obstruct the tube, with the C-terminal region of gpH forming a coiled-coil superhelix that interacts with gpI. The AHS and CSF domains within the tail fiber are twisted, and the bottom ends of the three RBDs are in close proximity (Fig. 6, step 1).

Upon recognition of LamB by the RBD of gpJ, the distance between the RBD bottom ends expands, effectively acting as a lever that transduces the upper side of the RBDs. The newly introduced hydrogen bonds between the upper side of RBDs and CSF induce CSF's rotation (Fig. 6, step 2). The degree of rotation increases as it propagates upwards, leading to the AHS flipping and partially transforming into β-sheets (Fig. 6, step 3). This twisting action results in the flipping and dissociation of AHS and the three strings of FNIII domains. The bottom end of the tail tube opens, with the FNIIIs moving to a position perpendicular to the TTC HD domains, similar to that in bacteriophage T5[27]. Subsequently, the HDs undergo significant positional changes. This induces the rigid body rotation of HDII-HDIII and HDII-ins1, along with the movement of HDII-ins2, collectively opening the tail tube. In sync with the movement of the HDs, the segment of gpI passing through them also moves in the same direction. The trimeric plug

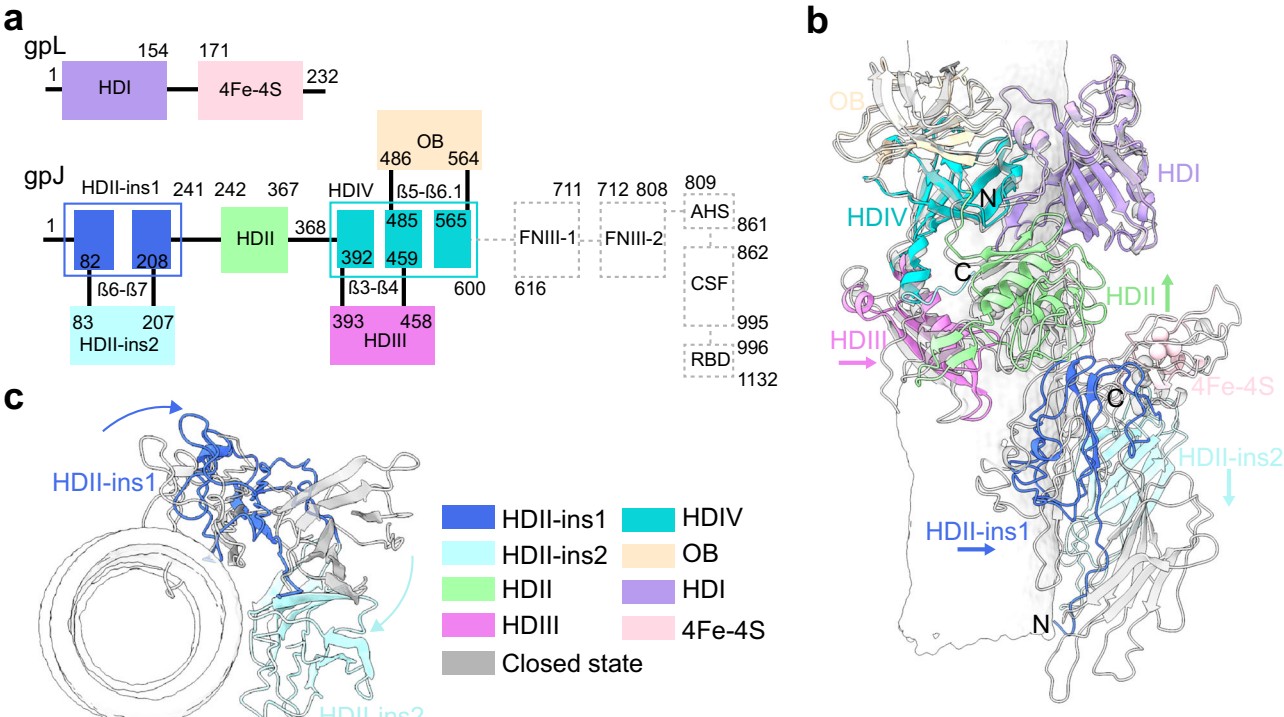

**Fig. 4 | Alterations in the gpJ and gpL HDs after the tail opening. a** The arrangement of sequence and domains of gpL and gpJ. The C-terminal FNIIIs, AHS, CSF, and RBD of gpJ are not included in **b. b** Depiction of structural variances in HDs between the closed (white-gray) and open states. Notable shifts are observed in HDII, HDIII, HDII-ins1, HDII-ins2 and 4Fe-4S. HDII, HDIII, HDII-ins1, and 4Fe-4S

undergo rigid body rotation. The potential DNA channel is illustrated by a surface, generated using HOLE software[45]. **c** Bottom view of **b**, highlighting an inward rotation of HDII-ins2 and a horizontal displacement of HDII-ins1, with other parts omitted for clarity.

domains of gpI disengage from each other. The upper portion of gpI and the plugs collapse towards the host cell's outer membrane. The two hydrophobic helices, H1 and H2, within gpI, potentially form a transmembrane channel by inserting into the outer membrane. This is facilitated by a previous significant reduction in the distance between gpI and the outer membrane, a result of the conformational change of AHS. In the open state, the inner diameter of the tail tube is sufficient for the C-terminus of gpH to pass through. Following the expulsion of gpH from the tail tube, viral DNA flows through the tube toward the inner membrane and finally enters the host cell for replication[28] (Fig. 6, step 4). Despite the insights gained, our study presents some limitations. The enzyme from the lambda phage that is responsible for cleaving the cell wall peptidoglycan in the periplasmic space remains unidentified, though it is speculated to possibly be the N-terminal part of gpI. Additionally, the mechanism by which lambda DNA traverses the periplasmic space and inner membrane is still unknown. Further molecular and structural studies are needed to elucidate these interactions and their implications for the phage life cycle.

These structures primarily affirm that, following the binding to the bacterial outer membrane receptor, the lambda tail undergoes a global structural alteration reminiscent of T5, involving the bending of the central tail fiber and a lateral descent and opening of the tube[22,27,29]. Both lambda phage and T5's FNIIIs are arranged in a way where one part moves to one side and two monomers move to the other, thereby orienting vertically to the tube. Similarly, the plugs in the closed tube structures of both lambda and T5 become loose and dangle in the open structures, suggesting a shared mechanism related to outer-membrane anchoring. Moreover, this study compares high-resolution structures of the central tail fiber in both closed and open states of lambda phage. It provides a detailed description of the changes in the AHS and CSF regions of Lambda phage upon receptor binding. Outer membrane receptors OmpF and OmpC,

similar to LamB, have β-barrels that are smaller than that of LamB (Supplementary Fig. S6a). Notably, clashes occur between the loop of OmpC and the open state of gpJ RBD, which are not observed with OmpF (Supplementary Fig. S6b). This observation, therefore, explains the discovery that the outward expansion at the bottom of RBD and the contraction at the top, triggered by its binding to LamB's ELs, serve as a trigger for the opening of the lambda phage tail. It also offers insight into why OmpF, rather than OmpC, can more readily become a receptor exploited by gpJ through point mutations[30-33]. For the first time, it is possible to illustrate the specific alterations across various regions of the central tail fiber triggered by binding to the outer membrane receptor, significantly enhancing our understanding of the signal transduction and structural changes in this region.

In summary, the implications of phage-receptor complex extend beyond the phage itself. The mechanisms of different bacteriophage-host interactions are central to many areas of microbiology and have potential applications in areas such as phage therapy, a promising alternative to antibiotics in the era of increasing antibiotic resistance[34]. The small changes in the RBD region and the substantial variations in the structures of AHS and FNIIIs in both open and closed states also illustrate a fascinating biological phenomenon.: how a minute binding event can trigger a series of structural changes and transductions within protein parts, thereby instigating a complex restructuring of a biological machine. This restructuring enables a molecular switch between two stable states, allowing the machine to carry out its complete biological function.

## Methods
### Purification of gpJ and MSP2N2
The recombinant plasmids encoding various gpJ constructs, each featuring an N-terminal 6xHis tag, were transformed into *E. coli* BL21

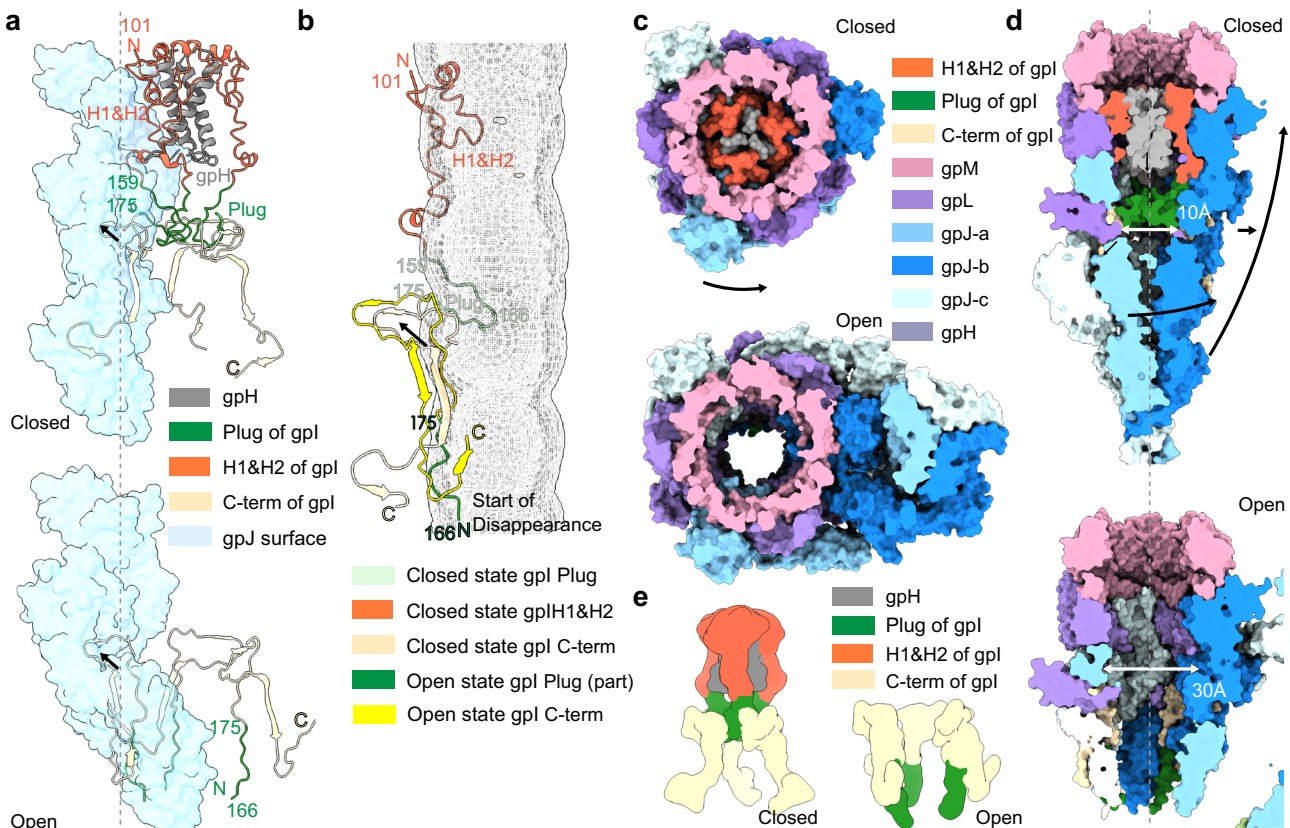

**Fig. 5 | Comparison of gpI protein and tail tip hole in closed and open states. a** Top: the gpI trimer (wheat, green and orange, cartoon), a portion of gpH trimer (gray) and a portion of gpJ monomer (blue, transparent surface) in the closed state. Bottom: the gpI (wheat and green, cartoon) and part of gpJ in the open state. Changes in the HD domains induce an outward shift of a gpI segment lodged within their interstices, and the gpI N-terminal above this space descends. The plug domain (green), which initially extends into the tip hole, is released. **b** The relative position of gpI after alignment of the tail tip complex with gpM. The predicted transmembrane entities H1 and H2 vanished from the lowest point of gpI in the open state. **c** Top view of the entire model surfaces. The plug domains of the gpI trimer disassemble in coordination with the rotation of the HD domains. **d** Representation of the central slice sideview of the entire model. Top: the closed state, the pore formed by the HD domains is 1 nm in diameter, tapering downwards until it is completely closed. In the open state, the narrowest part of the pore measures approximately 3 nm in diameter. Both the entire tip and the FNIII-1 and FNIII-2 of gpJ show noticeable positional changes compared to their locations in the closed state. **e** Schematic representation of gpI movement with HDs of gpJ (not shown).

(DE3) cells. Upon reaching an OD600 of 0.8 at 37 °C, protein expression was induced with 0.5 mM IPTG for 3 h.

The induced cells were collected via centrifugation and resuspended in buffer (25 mM Tris, 150 mM NaCl, pH 8.0). Cell disruption was performed using a French Press at 700–900 MPa for 4–6 cycles. The cellular debris was eliminated through centrifugation at 12,000 rpm for 60 min. The resulting supernatant was applied to a nickel gravity column. After washing with lysis buffer augmented with 25 mM imidazole, the protein was eluted with lysis buffer supplemented with 250 mM imidazole. The eluted proteins were concentrated through a 30 kDa cutoff centrifugal filter at 3000 × *g*, followed by gel filtration. The MSP2N2 protein was purified following the same protocol.

### Purification of LamB
LamB genes from various species were engineered by substituting the N-terminal signal peptide with the OMPG signal peptide and appending a Twin-Strep-tag post signal peptide. The modified plasmids were transformed into *E. coli* BL21 (DE3) cells. The cells were cultivated to an OD600 of 1.0 at 37 °C, then chilled to 18 °C, and protein expression was induced overnight with 0.5 mM IPTG.

The induced cells were gathered by centrifugation and resuspended in 15 mL of buffer (25 mM Tris, 150 mM NaCl, pH 8.0). Cell disruption was achieved using a French Press at 700–900 MPa for 4–6 cycles. The cell debris was cleared by centrifugation at

12,000 rpm for 10 min, and the supernatant was further clarified by centrifugation at 41,000 rpm for 1 h. The resulting pellet was resuspended in 4 mL of 20 mM Tris-HCl, pH 8.0 buffer supplemented with 2% N-Lauroylsarcosine sodium salt, and homogenization was carried out to extract the *E. coli* inner membrane proteins. The sample was then centrifuged at 41,000 rpm for 50 min at 4 °C, and the supernatant was discarded. The resultant *E. coli* outer membrane was resuspended in 2 mL of 20 mM Tris-HCl, pH 8.0 buffer containing 2% n-Dodecyl-B-D-Maltoside and homogenized to extract the E. coli outer membrane proteins. The sample was then incubated at 4 °C for 2 h. The resuspended *E. coli* outer membrane proteins were loaded onto a Strep-Tactin®XT gravity flow column and washed with 35 mL of 25 mM Tris-HCl, pH 8.0 buffer containing 0.02% DDM. The proteins were eluted with 25 mM Tris-HCl, pH 8.0 buffer containing 0.02% DDM, and 50 mM biotin. The eluted proteins were concentrated using a 50 kDa cutoff centrifugal filter at 2500 × *g*, followed by gel filtration.

### Tail purification
The expression plasmids carrying the tail assembly gene were transformed into BL21 (DE3) pLySs cells. The cells were cultured in LB medium containing ampicillin, chloramphenicol, and 5 mM glucose at 37 °C with a stirring speed of 220 rpm until the OD600 reached approximately 0.8. Expression was then induced by adding 1 mM IPTG for 4 h.

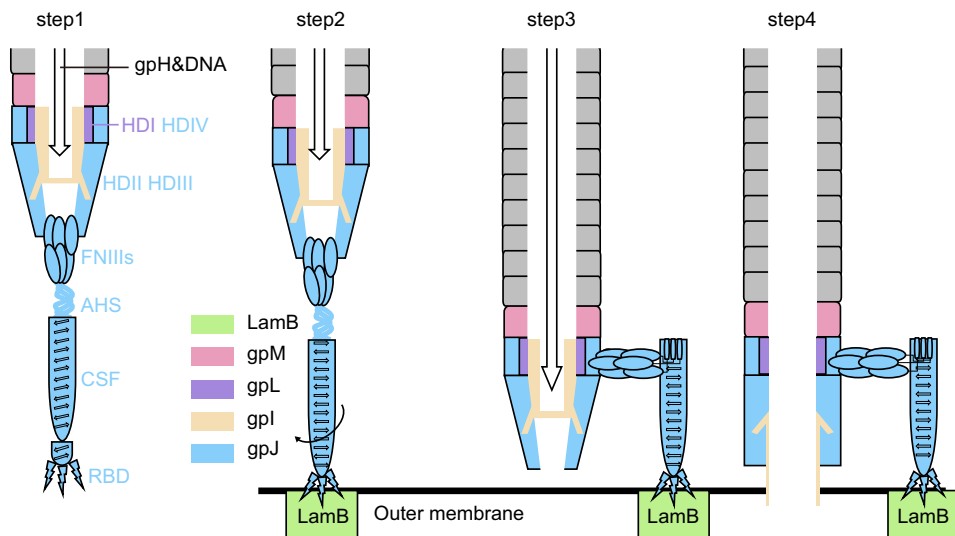

**Fig. 6 | Proposed mechanism for the conformational changes driven by receptor binding that triggers DNA ejection.** Step 1. The lambda tail in closed state. The HDs, gpI, and gpH in this state are referenced from PDB: 8IYK or 8IYL. Furthermore, there are newly solved models specifically for the closed state of AHS to RBD, which can be found in PDB: 8XCK. Step 2. Upon binding to LamB, the widening at RBD's base instigates an inward movement at the RBD apex. The genesis of new hydrogen bonds at CSF's base compacts its structure, inciting a rotation. Newly solved models are available for the open sate of RBD with LamB, which can be found in PDB: 8XCJ. Step 3. The rotation angle intensifies as it ascends. The AHS flips, subsequently wrapping around the top of CSF. Meanwhile, another portion undergoes transformation to form the β-sheet at the top of CSF. Three FNIII strings dissociate, two strings relocate on one side, and the third one on the other side. The distance between the other parts of the tail and outer membrane is reduced. Newly solved models are available for the open state of FNIII-1, FNIII-2, AHS and CSF, which can be found in PDB: 8XCI. Step 4. The HDII, HDIII, HDII-ins1, 4Fe-4S, and gpI located within the gaps of the HDs undergo a rigid body shift. The upper gpI descends, and the plug domains of gpI separate, resulting in the effective opening of the tubes. Newly solved models are available for the open state of HDs and gpI, which can be found in PDB: 8XCG.

The induced cells were harvested via centrifugation and the cell pellet was resuspended in 100 mL of buffer (25 mM Tris, 150 mM NaCl, pH 8.0) containing 2% Triton X-100, 2 mM PMSF, and 5 mM EDTA. DNaseI and 6 mM MgSO4 were added prior to the sonication of the cells. The cell debris was removed through centrifugation at 12,000 rpm for 14 min, and the resulting supernatant was collected. To this, 14.2% potassium glutamate and 7.5% PEG 8000 were added while stirring, and the mixture was chilled on ice for 30 min. This was followed by centrifugation at $14,400 \times g$ for 10 min, and the pellet was resuspended in 25 mL of TKG buffer (20 mM Tris-HCl, 100 mM potassium glutamate, pH 7.5 at 25 °C). After centrifuging at $20,200 \times g$ for 10 min, the supernatant was collected for density gradient centrifugation.

The sample was centrifuged at 29,000 rpm using an AH-629 rotor for 3.5 h in a 10−30% glycerol density gradient. The target band was collected and transferred to an ultracentrifuge tube, which was then centrifuged at 41,000 rpm for 2 h using an F50L-8×39 rotor. The supernatant was discarded, and the pellet was gently resuspended in 2 mL of TKG buffer. The sample was then applied to a Q column (SOURCE 15Q, 10/100 GL) and eluted with a gradient of 0−0.6 M salt. The elution fraction carrying the target protein was identified through negative staining. This fraction was concentrated using a 50 kDa cutoff centrifugal filter at $3000 \times g$, and then subjected to gel filtration using a Superose 6 Increase column (SR6).

### LamB in nanodisc
27 mg of *E. coli* polar lipid extract (Avanti) was dissolved in chloroform, and the solution was dried on the glass tube wall using nitrogen gas. To the dried lipids, 2 mL of 25 mM Tris-HCl, pH 8.0 buffer containing 14 mM DDM was added, and the mixture was sonicated until completely dissolved. The dissolved lipids were mixed with 12.4 mg of LamB protein and 20.5 mg of MSP2N2 protein (the final LamB/MSP/lipid ratio provided is 1:2.5:290) to reach a final volume of 10 mL, and the mixture was chilled on ice for 1 h. Biobeads (2 g) were added, and the mixture was incubated at 4 °C with rotation for 2 h to remove the detergent from the solution. The incubated complex was loaded onto a Strep-Tactin®XT gravity column for 3−5 cycles to remove excess MSP2N2 scaffold protein. The column was washed with 25 mM Tris-HCl (pH 8.0) buffer to remove any unbound protein. The target protein was eluted using a buffer containing 0.02% DDM, 25 mM Tris-HCl (pH 8.0), and 50 mM biotin, and then subjected to gel filtration using a Superose 6 Increase column (SR6).

### Tail-LamB complex purification
1.3 mg of the MSP2N2-LamB protein complex was mixed with 500 μL of the concentrated anion exchange purified bacteriophage tail complex, and the total volume was adjusted to 1050 μL. The mixture was gently mixed and left to stand at room temperature for 50 min. The unbound tail complex was removed using a nickel gravity column and the eluted sample was subjected to gel filtration using a Superose 6 Increase column (SR6). The elution peak containing the target protein complex was collected.

### gpJ-LamB complex in detergent
9 mg of LamB and 7 mg of gpJ713 were incubated at room temperature for an hour in lysis buffer containing 0.02% DDM. The unbound LamB protein was removed using a nickel gravity column, followed by the removal of unbound gpJ using a Strep-Tactin®XT gravity column. The sample was then subjected to size exclusion chromatography using a superdex column.

### Cryo-EM sample preparation
A 4 μl protein sample was placed on an Au 300 mesh Quantifoil R 1.2/1.3 EM grid that had been glow-discharged (Med, 30 s). The grid was then

rapidly frozen in liquid ethane that had been chilled with liquid nitrogen, using the Mark IV Vitrobot device from Thermo Fisher Scientific. The grid was blotted for 3 s after a 5 s waiting period, and then flash frozen in liquid ethane cooled by liquid nitrogen with a Vitrobot Mark IV (FEI) at 100% humidity and 8 °C.

## EM data acquisition
Data collection included 6811 images for the gpJ713, 2566 images for the gpJ713 and LamB complex, and 7714 images for the tail and LamB complex. Imaging of gpJ713-LamB complex and tail-LamB complex was performed using a 300 kV Titan Krios electron microscope (Thermo Fisher) outfitted with a K3 Summit counting camera (Gatan), with a pixel size of 1.0742. Imaging of gpJ713 was performed using a 300 kV Titan Krios electron microscope (Thermo Fisher) outfitted with a Falcon4 camera (Thermo Fisher), with a pixel size of 1.036. Each image stack received an approximate total dose of 50 e−/Å², with defocus values ranging between −1.5 and −2.5 μm. Data collection was fully automated, facilitated by AutoEMation software[35] and EPU (Thermo Fisher).

## EM image processing
Motion correction was performed using MotionCor2 v1.2.6[36], while GCTF v.1.18[37] was used for CTF estimation. The TsingTitan.py program, developed by Dr. Fan Yang, automatically executed these tasks using Gautomatch v.0.56 (https://www.mrc-lmb.cam.ac.uk/kzhang/Gautomatch/). CTF estimation was further carried out using CryoSPARC[38]. Micrographs with a CTF fitting resolution worse than 6 Å were discarded. All subsequent processing steps were conducted using cryoSPARC.

In the case of the tail-LamB complex, a trained Topaz model was utilized for automated particle picking, yielding 2,468,342 picked particles. These particles were further processed through bin4 extraction and 2D classification. From these, 358,235 particles were selected for ab-initio model generation. The best model was chosen for local refinement. The selected particles were re-extracted with shifts applied and then locally refined using different masks, resulting in two high-resolution maps focusing on distinct regions.

Post blob picking, inspection, extraction (box size of 192 pixels), and 2D classification, 62,334 particles from the gpJ713 and LamB complex were chosen. These were used to generate two ab-initio models in C1 symmetry. After homo-refinement and NU-refinement, the best map served as a reference for template picking across the entire dataset. In the end, 621,549 cleaned particles were obtained from 2D classification, and used to generate two ab-initio models in C1 symmetry. Following a 10-pixel shift of the particle centers and bin1 extraction with a box size of 216, NU-refinement was performed, resulting in a map at 2.98 Å resolution.

Post blob picking, inspection, extraction (box size of 192 pixels), and 2D classification, 1,378,512 particles from the gpJ713 were selected and used to generate two ab-initio models in C1 symmetry. After NU-refinement in C3 symmetry and center shifts, particles were bin1 extracted with a box size of 480, NU-refinement was performed, resulting in a map at 2.76 Å resolution.

## Protein model building and structure refinement
UCSF Chimera[39] was utilized to dock the gpM, gpL, tail tube domains, and FNIIIs domains of gpJ (PDB ID: 8IYK) into the cryo-EM map. The AHS, CSF, and RBD domains of gpJ were constructed de novo using EMBuilder[40]. Manual modifications to the models were conducted in COOT[41]. Structure refinements were subsequently executed in real space utilizing PHENIX[42]. Supplementary Table S1 contains comprehensive information regarding the 3D reconstruction and model refinement. All structural figures were generated using PyMol[43] and ChimeraX[44].

## Biolayer interferometry assay
The interaction between gpJ713 and various species of LamB proteins was characterized using biolayer interferometry (BLI) with Ni-NTA sensors. Initial equilibration of the sensors was performed in a buffer containing 25 mM HEPES-8.0 and 150 mM NaCl. Subsequently, the sensors were immersed in a 2 μg/mL solution of His-tagged gpJ713 protein for capture. Unbound gpJ713 was removed by equilibrating the sensors in lysis buffer with 0.02% DDM. The sensors were then immersed in various concentrations of Twin-Strep-tagged LamB proteins in lysis buffer with 0.02% DDM to evaluate the binding affinity. The dissociation kinetics were assessed by re-immersion of the sensors in lysis buffer with 0.02% DDM.

## Reporting summary
Further information on research design is available in the Nature Portfolio Reporting Summary linked to this article.

## Data availability
Cryo-EM maps and the associated structural coordinates have been respectively deposited into the Electron Microscopy Data Bank (EMDB) and the Protein Data Bank (PDB) under the following accession codes: EMD-38242/8XCG (Tail tube), EMD-38244/8XCI (Tail fiber), EMD-38245/8XCJ (gpJ713-LamB), and EMD-38246/8XCK (gpJ713). Source data are provided with this paper.

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

## Acknowledgements

We thank Tsinghua University Branch of China National Center for Protein Sciences (Beijing) for kindly providing the cryo-EM facility support and the computational facility support on the cluster of Bio-Computing Platform. We appreciate J. Lei and F. Yang for their technical support. This work was supported by grants from the National Natural Science Foundation of China [grant numbers 32371254 & 32171190].

## Author contributions

X.G. and J.W. conceived the project. X.G. prepared phage lambda tails, optimized cryo-grid preparation, recorded the cryo-EM data, and processed the cryo-EM data. J.W. built the atomic models. X.G. and J.W. wrote the manuscript.

## Competing interests

The authors declare no competing interests.
