## [Peer Review File · Nature Communications]

Structural mechanism of bacteriophage lambda tail's interaction with the bacterial receptorREVIEWER COMMENTS

Reviewer #1 (Remarks to the Author):

The paper from Ge and Wang present the structure of the tail tip of phage lambda in interaction with its receptor, LamB, inserted into a nanodisc. Comparing this structure with the structure of native lambda tail tip previously published, the authors propose a mechanism to explain the trigger for infection following phage interaction with its receptor.

Whereas this is a very interesting set of data, its originality and of that of the mechanism proposed are questionable, as they are very similar to the already published data concerning phage T5 (Linares et al, *Sci Adv*, 2023 and Degroux et al, *Virology*, 2023). Phage lambda and phage T5 belong to the same phage morphogroup, the siphoviridae.

Thus, the last phrase of the abstract stating that the work provides "novel perspectives on bacteriophage infection initiation" and "significantly advances the understanding of bacteriophage-bacterial interaction" is not correct and needs to be deleted.

Furthermore, for one that is not familiar with lambda tail, the paper is not clearly written, some descriptions are somewhat vague; the authors are not always consistent with their nomenclature and, from one paragraph to the other, call the same domain differently, which confuses the reader. Furthermore, the structure of the closed tail tube is not introduced, leaving the reader completely lost when invited to understand how the tail tube opens. The figures are not well annotated, it is difficult to relate domains noted in the text and structures in the figures. At least, N and C termini should be indicated, and domains better delimited. Also, as the authors are also co-authors of the paper describing lambda's tail before interaction with the receptor, and because they compare the two structures, they could use the same nomenclature (and colour code?) (eg. CSF vs BSP).

Major points

The authors could show a low resolution map of their structure of the tail tip in interaction with LamB in nanodisc: is a channel visible in the nanodisc, as expected if the proposed mechanism is correct?

The architecture of the paper is a bit awkward. The authors have chosen to present the interaction of the Receptor Binding Domains of gpJ with LamB and the induced conformational changes in the rest of gpJ before having introduced the structure of gpJ, which is rather confusing.

The authors describe structures and conformational changes in the text, refer to the figures, but it is sometimes difficult to relate text and figure. The figures should be more annotated to make the link. For example, line 145 "Additionally, its (the BSP) cross-section progressively expands from narrow to wide (Fig. 3A, left)." In Fig. 3A, left, where does BSP start, where does it end? Where is it narrow, where wide (from "bottom" to "top" or "top" to "bottom"?). Line 154: "The lower α -helix (residues 826-865) within the AHS remodels into a shorter α -helix segment, flanked by an extended two-layered β -sheet at the summit of the BSP (Fig. 3B)." It is absolutely impossible to understand what becomes what. This sequence could be highlighted in a different colour to allow the reader to see how it is in one and the other structure.

Lines 233-235: "The degree of rotation increases as it propagates upwards, leading to the α -helical shaft (AHS) flipping and partially transforming into β - sheets (Fig. 6, step 3)." This is not clear at all from the result section nor the figures, as indicated above. This is where the fibre bends?! This should be made much clearer in the results/figures.

The authors are sometimes vague. Please be more precise in the descriptions/explanations. For example:

Line 35: "Bacteriophage lambda, a member of the Siphoviridae family". Taxonomy has changed. Please update. Line 123: "Upon binding with LamB, the distance between the three protomers of RBDs increases from 1.8 nm to 3 nm (Fig. 2B)": which distance? If I understand well, it is increased at one extremity, but decreases on the other.

Line 163: "Upon ejection, the distance from the FNIII to the remote end of the RBD markedly shrinks from 29.7 nm to 13.2 nm." Upon ejection of what?! Do the authors mean ejection of DNA? However, the presented sample consists of purified tails, there is thus no DNA ejection.

From reading this manuscript, it is not clear at all how lambda's tail tube is closed. Is it by gpJ's FNIII domains (line 184), by HDII-ins2 (fig. S3), gpI (line 196)?? I am completely lost! This is a crucial information to understand the opening of the tail tube!!

Line 179: "When comparing the TTC structures of phage lambda in closed and open states, the most significant changes are observed in the structure of the HDs of gpJ." This is a strange statement, because gpI undergoes the most important conformational changes in this area of the map...

Line 185: "Upon gpJ binding with LamB, the FNIIIs dissociate and transit to a position perpendicular to the HDs, which results in TTC's switching to the open state." Why do FNIIIs dissociate? This is one of the fundamental steps of the mechanism, it should be detailed and explained!

The "plug" domain of gpI remains a mystery (lines 198, 209, Fig. 5 etc). What does it correspond to? Fig. 5A,B shows two domains for gpI (green and orange), which is the plug? Is it a subdomain of the green domain in Fig. 5?

And where would be the predicted peptidoglycan hydrolase domain? Figures should be clearer on the different domains of gpI

Line 204: Prefer "tube lumen" to "DNA channel"?

Lines 209-210: "The gpI segment above the gaps descends, inducing a loosening and dropping of the plugs.

Line 210: what is the TTC "end"? This is not clear at all: what are loosening and dropping? The fact that I don't see what the plug is does not help understanding.

Line 225: Introduce gpH!

Lines 227-230: "This closed state arrangement prepares the distal end of the tail fiber for locating and binding to the receptor, LamB, located on the host cell's outer membrane." Trivial?

Lines 235-236: "This twisting action results in the dissociation of the three strings of FNIII domains." Why? How?

Line 236: "The distal end of the tail tip opens,". Tail tip is rather vague. Be more precise: tail tube. Also, be consistent throughout the text: either use distal and proximal, or top and bottom, not a mixture of both.

Lines 238-239: "Subsequently, the interaction network between HDII-HDIII, HDII-ins1, and HDII-ins2 becomes destabilized." How? Why? Again, this is very vague!

Lines 247: "This action enables the C-terminus of gpH to pass through the transmembrane channel." Why? Lines 248-250: "Following the expulsion of gpH from the tail tube, viral DNA flows through the tube toward the inner membrane and finally enters the host cell for replication". What about peptidoglycan digestion? How is the inner-membrane passed?

Lines 252-254: "For instance, it merits consideration that the N-terminus of gpI may play a more complex role than it is currently understood. It may interact with other proteins to facilitate the crossing of the outer membrane." Why? Are there experimental evidence for this interesting hypothesis? Would it not suffice? Why not rather mention its predicted peptidoglycan hydrolase?! Where does that domain end up in the post-receptor interaction structure?

Line 258-9: "involving the bending of the central tail fiber and a lateral descent and opening of the tube (22,30)." -> refs 28, 30.

Lines 259-260: "Both lambda phage and T5's FNIIIs are arranged in a way where one part moves to one side and two parts move to the other," Monomer rather than "part"?

Lines 261-2: "The opening of the tube is universally realized by the rigid rotation of HDII and HDIII in both cases." Why "universally"? In phage T5, the opening of the tube is insured by the unfolding of a plug domain.

Lines 262_264: "Similarly, the plugs in the closed tube structures of both lambda and T5 become loose and dangle in the open structures, suggesting a shared mechanism related to membrane penetration." In phage T5, the plug domain unfolds to anchor the tail tube to the outer-membrane. It does not form the channel through it. This is proposed to be performed by the tape measure protein.

Lines 264-5: "Moreover, this study unveils high-resolution structures of the central tail fiber in both closed and open states within Siphoviridae phages." Again, this is not an unveiling, as such a structure has already been described previously.

Line 278: "In summary, the implications of this work extend beyond the phage lambda itself." Indeed, it has already been shown for phage T5!

Lines 281-2: "These structural differences also illustrate a fascinating biological phenomenon" What structural differences??

Figures and legends:

Figure 1: right panel: not very informative those grey blobs.

Figure 2E: "Sideview from the outer membrane (OM) plane, highlighting the interacting loops originating from the RBD (L1, L2, L3, L4) and LamB (EL4, EL6, EL9)" => "highlighting the interacting loops originating from one RBD (L1, L2, L3, L4) and a LamB monomer (EL4, EL6, EL9)"

Figure 3: A: Please note the residues for domain limits for all domains. Domain limits are not clear. Why does the AHS domain have different residue boundaries in the legend and in the figure?

"The C-terminus of helices (843-865) in the AHS domains have transformed into two additional β -sheets at the top of the BSP". Please note in a different colour, it is not clear at all.

B: Why are the two last sheets in red?

C: not clear. Redundant with scheme of figure 6?

Figure 4: The tail tube is a channel, not an active transporter of DNA. Please change throughout the text.

Figure 5: A,B: the different domains of gpI are not clear at all. Please label on the figure clearly. Having a similar panel as fig 3C would help understand how the tube opens and have a more global view.

C: "The plug domains of the gpI trimer disassemble in coordination with the rotation of the HD domains." This does not illustrate the figure. Why does that come in fig 5 when we are talking of this closing and opening mechanism since the beginning?!

Figure 6: Please label the domains mentioned in the text of the discussion describing the figure (AHS, BSP, FNIII...) on the figure! Also, it might be a bit too schematic: shouldn't there be two FNIII per gpJ monomer? What is the black arrow?

Supplementary figures:

Figure S1: too small...

Please, use a scheme to highlight what we are looking at. Use the same scale within a panel. What is the difference between the yellow and the grey maps? I guess the squared map is the one picked, from the gallery of 3D classes?

"An unsharpened map reveals subtle outlines of the receptor-binding domain (RBD) and LamB, and features remain identifiable despite the absence of sharpening." Where?

Figure S3: E: It is not clear what the 5.8Å distance refers to. Reading the legend, I expected it to be from gpI to gpJ in the closed and open states, but the second arrow rather shows gpI closed to open...

The different domains of gpI are not clear.

Material and Methods:

What is the final LamB/MSP/lipid ration in mol?

Reviewer #2 (Remarks to the Author):

In this manuscript, Ge and Wang present EM structures of the tip of the bacteriophage lambda tail before and after binding to the lambda receptor LamB. This work is of high importance for phage biology and virus research in general (lambda is a very important model virus). The results are of high technical quality. They describe the structures in detail and provide important information about the necessary conformational changes, finishing with a schematic model for infection. The results are very interesting, but I would like to suggest some improvements in the presentation.

1. The work by Breyton et al on another siphovirus (T5) is mentioned and partially referenced, but a detailed comparison is missing. Their important Linares et al, Sci Adv paper is also not cited. I think a detailed discussion of the differences and similarities in the two works is very important.

2. In figure 6 and some earlier figures, it would be useful to include which parts of which steps have now exactly been structurally resolved, i.e. mention the PDB codes in the legends and the new resolved structures boxed.

3. Supplementary movie file(s) of the proposed conformational changes would be very helpful.

4. The description of the structures and conformation changes is a bit dense - I know that it is difficult to write these explanations in a way that is easy to follow, but some improvements could probably be made.

5. 8XCK-gpJ713 seems to have peptidylprolyl isomerase co-purified and bound in the structure (as identified by BLAST analysis of the sequence) - however I could not find any comment about that in the manuscript or supplementary material. The map also has density at the N-termini that was not modelled - could this be done?

6. Fig 3A shows a bigger structure than provided in PDB file 8XCK, where is this structure based on? Mention reference and PDB code. Fig 3C is unclear to me.

Minor points:

43: absent should be absence

111: not sure what is meant by "demonstrates a lateral descent", I suggest rephrasing.

115: The receptor binding domain...

Sonnei should not be capitalized, should be sonnei. In two places it is written Sonnei.

482: I would include "nanodisc" somewhere in the title of this section

ref 2: title does not need to be capitalised throughout

Fig 3: outter should be outer

Response to Referees' Comments

Referee #1:

The paper from Ge and Wang present the structure of the tail tip of phage lambda in interaction with its receptor, LamB, inserted into a nanodisc. Comparing this structure with the structure of native lambda tail tip previously published, the authors propose a mechanism to explain the trigger for infection following phage interaction with its receptor.

Whereas this is a very interesting set of data, its originality and of that of the mechanism proposed are questionable, as they are very similar to the already published data concerning phage T5 (Linares et al, Sci Adv, 2023 and Degroux et al, Virol J, 2023). Phage lambda and phage T5 belong to the same phage morphogroup, the siphoviridae. Thus, the last phrase of the abstract stating that the work provides "novel perspectives on bacteriophage infection initiation" and "significantly advances the understanding of bacteriophage-bacterial interaction" is not correct and needs to be deleted.

We sincerely appreciate your insightful and valuable comments. We genuinely appreciate your recognition of the intriguing nature of our data. We fully acknowledge the concern raised regarding the originality of our work and the proposed mechanism, as they share a substantial conceptual alignment with the previously published data on phage T5 (Linares et al, Sci Adv, 2023 and Degroux et al, Virol J, 2023).

We would like to emphasize that our study primarily aims to provide detailed structures of bacteriophage lambda infection initiation, which contributes to the ongoing advancements in the understanding of lambda-bacterial interaction. Taking into careful consideration your valuable comment, we have revised the last phrase of the abstract accordingly. The updated abstract now reads as follows: "Providing detailed structures of bacteriophage lambda infection initiation, this study contributes to the expanding knowledge of lambda-bacterial interaction, which holds significance in the fields of microbiology and therapeutic development."

Furthermore, for one that is not familiar with lambda tail, the paper is not clearly written, some descriptions are somewhat vague; the authors are not always consistent with their nomenclature and, from one paragraph to the other, call the same domain differently, which confuses the reader. Furthermore, the structure of the closed tail tube is not introduced, leaving the reader completely lost when invited to understand how the tail tube opens. The figures are not well annotated, it is difficult to relate domains noted in the text and structures in the figures. At least, N and C termini should be indicated, and domains better delimited. Also, as the authors are also co-authors of the paper describing lambda's tail before interaction with the receptor, and because they compare the two structures, they could use the same nomenclature (and colour code?) (eg. CSF vs BSP).

Thank you for your feedback. We sincerely appreciate your valuable suggestions, and we have made the following revisions based on your comments.

First and foremost, we would like to express our gratitude for your input regarding the descriptions. We have carefully reviewed the manuscript and made the necessary adjustments to ensure clarity and precision in our explanations. We have taken great care to maintain consistent nomenclature throughout the paragraphs, addressing any discrepancies to avoid confusion for the readers.

In addition, we have taken your advice into account and have added a more comprehensive introduction to the closed tail tube. This addition aims to assist readers who may not be familiar with the lambda tail in understanding the process of how the tail tube opens. We believe that this clarification will alleviate any confusion and help readers grasp the concept more effectively.

Furthermore, we have improved the annotations and labels in the figures to facilitate the correlation between the mentioned domains in the text and the corresponding structures in the figures. We have specifically marked the N-terminus and C-terminus and provided clearer demarcations for the domains.

Regarding the naming issue of CSF, we appreciate your concern about potential confusion caused by changing the name. Therefore, we have reverted to using CSF to maintain consistency with the previous publication describing the lambda tail structure (Wang *et al*, Structure, 2024). As for the color code, we understand that inconsistency can impede understanding. To address this, we have included a closed structure in the supplementary figure (Fig. S3) that aligns with the open state discussed in this study, enabling readers to better comprehend the relationship between the structures.

Once again, we sincerely appreciate your valuable suggestions, and we hope that our revisions address your concerns and improve the manuscript accordingly.

Major points

The authors could show a low resolution map of their structure of the tail tip in interaction with LamB in nanodisc: is a channel visible in the nanodisc, as expected if the proposed mechanism is correct?

We appreciate your question regarding the low-resolution map of the tail tip in interaction with LamB in nanodisc. We can indeed provide you with the relevant structure map and model as requested. However, it is important to note that we have not observed a visible channel in the nanodisc. This could be due to the challenges of creating a stable lipid environment with sufficient space for channel insertion while fully encapsulating the trimeric LamB receptor, even with the use of MSP2N2, which has a relatively large inner diameter. Despite this, we have observed some alignment between the positions of the tube and nanodisc in a subset of particles after performing 3D classification (Figure 1).

Figure 1 A low resolution map of the tail tip in interaction with LamB in nanodisc.

The architecture of the paper is a bit awkward. The authors have chosen to present the interaction of the Receptor Binding Domains of gpJ with LamB and the induced conformational changes in the rest of gpJ before having introduced the structure of gpJ, which is rather confusing.

You raise a valid point, and we sincerely appreciate your insightful observation. We acknowledge that introducing the structure of gpJ before presenting the interaction of the RBDs of gpJ with LamB and the induced conformational changes would have provided a clearer and more logical flow. In light of this feedback, we have taken appropriate measures to rectify the issue. Specifically, we have included an additional supplementary figure (Fig. S3) that illustrates the closed tail tip (includes gpJ), aiming to provide readers with a better understanding of the context.

The authors describe structures and conformational changes in the text, refer to the figures, but it is sometimes difficult to relate text and figure. The figures should be more annotated to make the link.

Thank you for your suggestion. We have made improvements to the annotations in the figures to ensure better alignment with the corresponding descriptions in the text. By adding more detailed labels, captions, and annotations, we aim to enhance the clarity and coherence between the figures and the accompanying text. We appreciate your valuable input and hope that these modifications address your concerns.

For example, line 145 “Additionally, its (the BSP) cross-section progressively expands from narrow to wide (Fig. 3A, left).” In Fig. 3A, left, where does BSP start, where does it end? Where is it narrow, where wide (from “bottom” to “top” or “top” to “bottom”?).

Thank you for providing specific feedback. We have taken your suggestion into account and made the necessary modifications. Firstly, we have changed "BSP" to "CSF" to align with the terminology used in the previous paper. Additionally, in Figure 3A, we have included labels indicating the starting and ending positions of the CSF region. To clarify the direction of expansion, we have revised the corresponding text to read: "From bottom side to top side, its cross-section progressively expands from narrow to wide (Fig. 3A, left)." These changes address your concerns and aim to provide clearer information regarding the CSF region in relation to the figure.

Line 154: “The lower α -helix (residues 826-865) within the AHS remodels into a shorter α -helix segment, flanked by an extended two-layered β -sheet at the summit of the BSP (Fig. 3B).” It is absolutely impossible to understand what becomes what. This sequence could be highlighted in a different colour to allow the reader to see how it is in one and the other structure.

Thank you for your suggestion. We appreciate your feedback, and we have made revisions to address the issue. In Figure 3, we have modified the representation by highlighting the corresponding regions of one chain in different colors in panel A and panel B. This visual distinction will allow readers to easily compare and understand the structural changes. Additionally, we have included a supplementary movie file that demonstrates the structural transitions in this specific region. We believe that these modifications will greatly enhance the clarity and improve the understanding of the remodeling process.

Lines 233-235: “The degree of rotation increases as it propagates upwards, leading to the α -helical shaft (AHS) flipping and partially transforming into β - sheets (Fig. 6, step 3).” This is not clear at all from the result section nor the figures, as indicated above. This is where the fibre bends?! This should be made much clearer in the results/figures.

Thank you for bringing this to our attention. We apologize for the lack of clarity in the result section and figures regarding the fiber bending and the transformation of the α -helical shaft (AHS) into β -sheets. We agree that this part should be made much clearer for readers to understand.

In response to your feedback, we have implemented your previous suggestion and highlighted the specific regions undergoing transformation in the figures 3A-B. This visual distinction will help readers easily identify the changes taking place. Additionally, we have included a supplementary movie to provide a more comprehensive and dynamic representation of these results.

The authors are sometimes vague. Please be more precise in the descriptions/explanations. For example:

Line 35: "Bacteriophage lambda, a member of the Siphoviridae family". Taxonomy has changed. Please update.

Thank you for bringing this to our attention. The phage family taxonomy has been revised according to Turner et al (doi.org/10.3390/v13030506) as follows: "Bacteriophage lambda, a member of the siphoviruses."

Line 123: "Upon binding with LamB, the distance between the three protomers of RBDs increases from 1.8 nm to 3 nm (Fig. 2B)": which distance? If I understand well, it is increased at one extremity, but decreases on the other.

Thank you for pointing out the ambiguity in the statement. You are correct in understanding that the distance between the three protomers of RBDs changes asymmetrically upon binding with LamB. To clarify this, we have revised the sentence to read: "Upon binding with LamB, the distance between the three protomers of RBDs on the bottom side increases from 1.8 nm to 3 nm (Fig. 2B)." This modification aims to provide a clearer description of the specific distance that undergoes an increase upon binding.

Line 163: "Upon ejection, the distance from the FNIII to the remote end of the RBD markedly shrinks from 29.7 nm to 13.2 nm." Upon ejection of what?! Do the authors mean ejection of DNA? However, the presented sample consists of purified tails, there is thus no DNA ejection.

Thank you for bringing this to our attention. You are correct in pointing out the inconsistency in the use of the term "ejection" in this context, as there is no DNA ejection in the presented sample. We apologize for the confusion caused.

To address this issue, we have revised the sentence as follows: "Compare to the closed state structure, the distance from the FNIII to the remote end of the RBD markedly

shrinks from 29.7 nm to 13.2 nm." This modification provides a more accurate description of the observed distance change without implying the ejection of DNA.

From reading this manuscript, it is not clear at all how lambda's tail tube is closed. Is it by gpJ's FNIII domains (line 184), by HDII-ins2 (fig. S3), gpl (line 196)?? I am completely lost! This is a crucial information to understand the opening of the tail tube!!

Thank you for your insightful comment. We appreciate your careful examination of our manuscript. In response to your query regarding the closure of lambda's tail tube, we acknowledge the lack of clarity in our initial presentation.

In consideration of your suggestion, we have revised the manuscript to include a detailed structural description of the closed state of the tube in the main text (the first sentence in related section) and corresponding figures in the appendix (Fig. S3). We believe that these additions will provide a clearer understanding of how the lambda tail tube is closed and help elucidate the subsequent changes upon its opening.

Line 179: "When comparing the TTC structures of phage lambda in closed and open states, the most significant changes are observed in the structure of the HDs of gpJ." This is a strange statement, because gpl undergoes the most important conformational changes in this area of the map...

We appreciate your keen observation and constructive feedback. We acknowledge that the initial statement in line 179 may have caused confusion.

In response to your suggestion, we have revised the sentence to more accurately reflect the structural changes in the tail tip complex (TTC) of phage lambda. The modified statement now reads: "When comparing the TTC structures of phage lambda in closed and open states, significant changes are observed in the structure of the HDs of gpJ."

Line 185: "Upon gpJ binding with LamB, the FNIIIs dissociate and transit to a position perpendicular to the HDs, which results in TTC's switching to the open state." Why do FNIIIs dissociate? This is one of the fundamental steps of the mechanism, it should be detailed and explained!

We appreciate your insightful comment. Concerning the reasons for FNIIIs dissociation, we have carefully considered the structural changes induced by receptor binding in the CSF and AHS regions. In the closed structure, the FNIII trimer's bottoms are tightened by a helix bundle of AHS. Upon binding with the receptor, structural changes occur in the CSF and AHS regions: a portion of the helix bundle transforms into β -sheets, and the remaining part wraps around the upper surface of CSF, resulting in an increased distance between AHS monomers. This transition releases constraints on the FNIII trimer's bottom side, allowing it to undergo dissociation.

We have revised the corresponding passage in the manuscript to reflect these structural insights: "Upon the binding of gpJ with LamB, the helix bundle of AHS undergoes a transition, forming a wrap around the upper surface of CSF. This transformation leads to a reduction in the restraints on the FNIII trimer from the bottom side. Subsequently, the FNIIIs dissociate and reposition themselves perpendicular to the HDs, inducing the transition of TTC to an open state."

We believe this modification provides a more comprehensive and elucidating description of the fundamental steps involved in the mechanism. Thank you for guiding us toward a more accurate representation of our findings.

The "plug" domain of gpI remains a mystery (lines 198, 209, Fig. 5 etc). What does it correspond to? Fig. 5A,B shows two domains for gpI (green and orange), which is the plug? Is it a subdomain of the green domain in Fig. 5?

Thank you for pointing out the ambiguity regarding the "plug" domain of gpI. We have updated Figure 5 and labeled the corresponding residue range (159 to 175) when introducing the term "plugs" in the main text.

And where would be the predicted peptidoglycan hydrolase domain? Figures should be clearer on the different domains of gpl.

Thank you for your comments. Based on the prediction made by AlphaFold2, it is evident that the protein gpK possesses the characteristic of being a cell wall-associated hydrolase. However, our analysis of the closed or open structure of the lambda tail does not indicate the presence of gpK. The protein gpl contains 223 residues, and our structural model of gpl encompasses only the C-terminal half (residues V102-R223). Due to the collapse of helices of H1 & H2 in the open state of gpl, the N-terminal domain of gpl appears to have fallen into the periplasmic space. As of now, there is insufficient evidence to definitively confirm whether the N-terminal domain of gpl acts as a peptidoglycan hydrolase. To exercise caution, we have opted to temporarily exclude all statements pertaining to the peptidoglycan hydrolase domain of gpl, acknowledging the lack of sufficient evidence to support these predictions.

Line 204: Prefer "tube lumen" to "DNA channel"?

Thank you for your suggestion regarding the terminology. We appreciate your input and have made the change from "DNA channel" to "tube lumen".

Lines 209-210: "The gpl segment above the gaps descends, inducing a loosening and dropping of the plugs. Line 210: what is the TTC "end"? This is not clear at all: what are loosening and dropping? The fact that I don't see what the plug is does not help understanding.

Thank you for your continued feedback and clarification. We appreciate your insightful comments. The imprecise term "the TTC 'end'" in line 210 has been revised to "the bottom of the tail tube" for better accuracy.

In the closed structure, the orange part is evident, while in the open structure, its absence suggests a shift of H1 and H2 closer to the outer membrane, deduced by analyzing the position of residue 116.

The plug's location has been explicitly marked in the updated figures and text. In the closed state, this region obstructs the tube, while in the open structure, it undergoes loosening. The top view in Figure 5C illustrates this change, demonstrating that the region no longer blocks the tube.

We hope these adjustments accurately convey the structural variations and address your concerns effectively.

Line 225: Introduce gpH!

Thank you for your suggestion. We have included a description of the trimeric gpH C-terminal helical bundle in gpl section and depicted its positioning in panel 5A, where it can be observed that gpH is blocked by the plugs of gpl above it.

Lines 227-230: "This closed state arrangement prepares the distal end of the tail fiber for locating and binding to the receptor, LamB, located on the host cell's outer membrane." Trivial?

Thank you for your comment. We acknowledge your concern about the perceived triviality of the statement in lines 227-230. Upon careful consideration, we have decided to remove the mentioned statement in lines 227-230.

Lines 235-236: "This twisting action results in the dissociation of the three strings of FNIII domains." Why? How?

Thank you for your question regarding the mechanism behind the dissociation of the three strings of FNIII domains. We appreciate the opportunity to provide a more detailed explanation.

The twisting action mentioned in lines 235-236 is a result of the rotation of the CSF. This rotation ultimately triggers the unfolding and flipping of the helix bundle in the AHS. During this process, a portion of the AHS transforms into β -sheets and wraps around the upper surface of CSF. This structural transition leads to an increase in the distance between three helices which connected to FNIIIs, potentially weakening the interactions at the bottom of the FNIII trimer.

As a consequence, the weakened interactions induce the dissociation of the three strings of FNIII domains. This explanation has been integrated into the revised manuscript, specifically in the Results section: "The loosen of helix bundle leads to a reduction in the restraints on the FNIII trimer from the bottom side,"

Furthermore, in the Discussion section, we have adjusted the description to align with these structural insights: "This twisting action results in the flipping and dissociation of AHS and the three strings of FNIII domains."

We hope these modifications provide a clearer understanding of the structural changes leading to FNIII dissociation.

Line 236: "The distal end of the tail tip opens,". Tail tip is rather vague. Be more precise: tail tube. Also, be consistent throughout the text: either use distal and proximal, or top and bottom, not a mixture of both.

Thank you for your observation regarding the clarity and consistency of our terminology. We appreciate your feedback. Upon review, we have made the necessary adjustments to ensure uniformity in our use of terms.

Specifically, we have replaced "The distal end of the tail tip opens," with "The bottom end of the tail tube opens," for precision. Additionally, we have replaced instances of "distal" and "proximal" with "top" and "bottom" to maintain consistency throughout the text.

Lines 238-230: "Subsequently, the interaction network between HDII-HDIII, HDII-ins1, and HDII-ins2 becomes destabilized." How? Why? Again, this is very vague!

Thank you for your valuable input. We appreciate your feedback. We have revised the sentence to provide a more explicit description: "Subsequently, the HDs underwent significant positional changes."

Lines 247: "This action enables the C-terminus of gpH to pass through the transmembrane channel." Why?

Thank you for your inquiry regarding the mechanism behind the C-terminus of gpH passing through the transmembrane channel. We appreciate your attention to detail.

In response, we have modified the sentence to provide a more specific explanation: "In the open state, the inner diameter of the tail tube is sufficient for the C-terminus of gpH to pass through."

We trust this adjustment offers a clearer understanding of the process, and we appreciate your diligence in improving the precision of our manuscript.

Lines 248-250: "Following the expulsion of gpH from the tail tube, viral DNA flows through the tube toward the inner membrane and finally enters the host cell for replication". What about peptidoglycan digestion? How is the inner-membrane passed?

Thank you for raising important questions about peptidoglycan digestion and the mechanism of passing through the inner membrane following the expulsion of gpH from the tail tube.

In response to your inquiry, we must acknowledge that our current structural analyses, including the examination of four different maps and the analysis of the previously published closed structure, do not provide sufficient insights to propose a detailed model for these processes. We recognize the limitations in our current understanding.

We appreciate your thoughtful comments and recognize the importance of addressing these aspects. We hope that future research endeavors will offer more comprehensive insights into the mechanisms of peptidoglycan digestion and the process of viral DNA passing through the inner membrane. We remain committed to advancing our understanding in these areas as the field progresses.

Thank you for your understanding and guidance.

Lines 252-254: "For instance, it merits consideration that the N-terminus of gpI may play a more complex role than it is currently understood. It may interact with other proteins to facilitate the crossing of the outer membrane." Why? Are there experimental evidence for this interesting hypothesis? Would it not suffice? Why not rather mention its predicted peptidoglycan hydrolase?! Where does that domain end up in the post-receptor interaction structure?

Thank you for your insightful comments regarding the speculation about the N-terminus of gpl and its potential role in interacting with other proteins for facilitating the crossing of the outer membrane. We appreciate your scrutiny of our manuscript.

Upon careful consideration and in alignment with your concerns, we acknowledge that the statement lacked sufficient experimental data to support this hypothesis. However, we also recognize the limitations in the credibility of the predicted peptidoglycan hydrolase domain.

In response to your feedback, we have removed all content related to the predicted peptidoglycan hydrolase and the speculative statement in lines 252-254. We aim to avoid overinterpretation of our structural findings and maintain a cautious approach in our conclusions.

Line 258-9: "involving the bending of the central tail fiber and a lateral descent and opening of the tube (22,30)." -> refs 28, 30.

We apologize for the citation error in lines 258-259. The correct references for the statement "involving the bending of the central tail fiber and a lateral descent and opening of the tube" are refs 28 and 30. We appreciate your attention to detail.

Lines 259-260: "Both lambda phage and T5's FNIII are arranged in a way where one part moves to one side and two parts move to the other," Monomer rather than "part"?

Thank you for pointing out the correction needed in lines 259-260. We appreciate your suggestion, and the term "part" has been appropriately replaced with "monomer" to enhance accuracy.

Lines 261-2: "The opening of the tube is universally realized by the rigid rotation of HDII and HDIII in both cases." Why "universally"? In phage T5, the opening of the tube is insured by the unfolding of a plug domain.

Thank you for your observation regarding the term "universally" in lines 261-262. We appreciate your clarification, and in response to your feedback, the sentence has been removed from the manuscript.

Lines 262_264: "Similarly, the plugs in the closed tube structures of both lambda and T5 become loose and dangle in the open structures, suggesting a shared mechanism related to membrane penetration." In phage T5, the plug domain unfolds to anchor the tail tube to the outer-membrane. It does not form the channel through it. This is proposed to be performed by the tape measure protein.

Thank you for providing further clarity on the role of the plug domain in phage T5 and its proposed mechanism. We appreciate your insights.

In response to your valuable information, we have revised the statement to accurately reflect the distinction: "Similarly, the plugs in the closed tube structures of both lambda and T5 become loose and dangle in the open structures, suggesting a shared mechanism related to outer-membrane anchoring."

We hope this modification better aligns with the observed mechanisms in both phages.

Lines 264-5: "Moreover, this study unveils high-resolution structures of the central tail fiber in both closed and open states within Siphoviridae phages." Again, this is not an unveiling, as such a structure has already been described previously.

Thank you for your observation regarding the term "unveiling" and the need for more accurate language. We appreciate your feedback.

In response, we have modified the statement to better reflect the contribution of the study: "Moreover, this study compares high-resolution structures of the central tail fiber in both closed and open states of lambda phage. It provides a detailed description of the changes in the AHS and CSF regions of Lambda phage upon receptor binding."

We believe this revision more accurately conveys the focus and contribution of the study. Thank you for guiding us toward a clearer expression.

Line 278: "In summary, the implications of this work extend beyond the phage lambda itself." Indeed, it has already been shown for phage T5!

Thank you for your insightful comment regarding the implications of the work and the importance of acknowledging the foundational contribution of the phage T5 study.

In response, we have modified the statement to better reflect the intended emphasis on the broader applications of phage-receptor complexes: "In summary, the implications of phage-receptor complex extend beyond the phage itself. The mechanisms of different bacteriophage-host interactions are central to many areas of microbiology and have potential applications in areas such as phage therapy, a promising alternative to antibiotics in the era of increasing antibiotic resistance."

We hope this revision appropriately highlights the broader significance of phage-host structure study.

Lines 281-2: "These structural differences also illustrate a fascinating biological phenomenon" What structural differences??

Thank you for your keen observation and feedback. We appreciate your guidance in providing more specific language to describe the structural differences.

In response, we have revised the statement to address your concern: "The small changes in the RBD region and the substantial variations in the structures of AHS and FNIIIs in both open and closed states also illustrate a fascinating biological phenomenon."

We hope this modification accurately captures the intended meaning.

Figures and legends:

Figure 1: right panel: not very informative those grey blobs.

Thank you for your feedback regarding the right panel of Figure 1. We appreciate your comment on the lack of informativeness with the grey blobs. In response to your observation, we have replaced the original representation with a low-pass filtered schematic, consistent with the color code used in other figures. This modification aims to enhance the reader's understanding of the overall changes in the phage tail.

We hope this revision aligns better with your expectations, and we are grateful for your valuable input.

Figure 2E: "Sideview from the outer membrane (OM) plane, highlighting the interacting loops originating from the RBD (L1, L2, L3, L4) and LamB (EL4, EL6, EL9)" => "highlighting the interacting loops originating from one RBD (L1, L2, L3, L4) and a LamB

monomer (EL4, EL6, EL9)”

Thank you for your suggestion. The caption for Figure 2E has been succinctly revised: "Sideview from the outer membrane (OM) plane, highlighting interacting loops from one RBD (L1, L2, L3, L4) and a LamB monomer (EL4, EL6, EL9)." We appreciate your concise feedback.

Figure 3: A: Please note the residues for domain limits for all domains. Domain limits are not clear. Why does the AHS domain have different residue boundaries in the legend and in the figure?

Thank you for bringing attention to the clarity of domain limits in Figure 3A. We appreciate your feedback.

In response, we have added an overview of the overall closed structure of gpJ, and we have ensured uniform and consistent labeling of the boundaries for each domain. This adjustment aims to enhance the clarity and accuracy of the figure.

“The C-terminus of helices (843-865) in the AHS domains have transformed into two additional β -sheets at the top of the BSP”. Please note in a different colour, it is not clear at all.

Thank you for your feedback regarding the clarity of the statement about the C-terminus of helices in the AHS domains in Figure 3. We appreciate your suggestion.

In response, we have changed the color to highlight this specific region in the figure. We hope this modification enhances the understanding for readers.

B: Why are the two last sheets in red?

Thank you for your inquiry. The two last sheets were highlighted in red to emphasize the last two beta-sheets in the CSF (BSP) region, which undergo significant changes as mentioned in Figure 2. To avoid any confusion, we have now changed the color of these sheets to black.

C: not clear. Redundant with scheme of figure 6?

Thank you for your feedback on Figure 3C. We retained this panel with the intention of providing a low-pass surface representation to assist readers unfamiliar with phage tail structures in gaining a macroscopic understanding of the rotation in the CSF region and the changes in the AHS region. In addition, we have included a corresponding movie to further aid readers in comprehending this structural aspect.

We hope that by offering both the visual representation and the accompanying movie, the information is presented more comprehensibly to a broader audience. If you have any further suggestions or specific concerns, we are open to additional modifications.

Figure 4: The tail tube is a channel, not an active transporter of DNA. Please change throughout the text.

Thank you for pointing out the clarification needed regarding the tail tube. We have made the necessary corrections throughout the text to accurately describe it as a channel rather than an active transporter of DNA.

Figure 5: A,B: the different domains of gpl are not clear at all. Please label on the figure clearly.

Thank you for your constructive feedback regarding Figure 5A and 5B. We sincerely appreciate your insightful input, and in response, we have diligently improved the figures by providing clearer annotations for each domain, including the plug domain.

Having a similar panel as fig 3C would help understand how the tube opens and have a more global view.

Thank you for your valuable suggestion. We appreciate your insight. In response to your feedback, we have added a low-pass surface representation in Figure 5E, similar to that in Figure 3C. This addition aims to enhance the understanding of the structural changes in gpI, providing a more global view of how the tube opens. We hope this modification aligns with your expectations, and we appreciate your thorough review.

C: "The plug domains of the gpl trimer disassemble in coordination with the rotation of the HD domains." This does not illustrate the figure. Why does that come in fig 5 when we are talking of this closing and opening mechanism since the beginning?!

Thank you for your valuable feedback. We understand your concern about the placement of the information regarding the disassembly of plug domains in Figure 5, and we appreciate the opportunity to provide clarification.

The inclusion of this information at this point in the figure is intended to highlight the correlation between the displacement of HD domains and the disassembly of plug domains. We aim to illustrate how the structural changes in HDs lead to a coordinated movement of gpl, impacting the plug domains. To enhance clarity, we have incorporated a representation in panel A to visually depict the changes in HDs and the corresponding movement of gpl. Additionally, the same theme is continued in Supplementary Figure 3E.

This discussion is positioned in Figure 5 to systematically compare the structural changes in each region, starting from the receptor-binding region. This strategic placement is intended to facilitate a better understanding of the process for readers less familiar with phage tail structures.

We hope this explanation clarifies the rationale behind the placement of this information in Figure 5. If you have further suggestions or if there is a preferred approach, we are open to modifying the presentation to better suit your expectations.

Figure 6: Please label the domains mentioned in the text of the discussion describing the figure (AHS, BSP, FNIII...) on the figure! Also, it might be a bit too schematic: shouldn't there be two FNIII per gpJ monomer? What is the black arrow?

Thank you for your valuable feedback regarding Figure 6. We appreciate your suggestions and have made the following modifications:

- Added labels for the mentioned domains (AHS, BSP, FNIII) on the figure.
- Revised the representation to include two FNIII per gpJ monomer.
- Black arrow means gpH&DNA (label added).

These adjustments aim to enhance clarity and better align the figure with the text discussion.

Supplementary figures:

Figure S1: too small...

Thank you for bringing attention to the size issue in Figure S1. Recognizing this concern, we have divided the structural calculation flowchart into two separate figures, S1 and S2, to ensure better visibility and clarity. Your feedback has been instrumental in improving the presentation of our work.

Please, use a scheme to highlight what we are looking at. Use the same scale within a panel. What is the difference between the yellow and the grey maps? I guess the squared map is the one picked, from the gallery of 3D classes?

Thank you for your valuable suggestions. We have reorganized the layout of the flowchart, ensuring consistent size scale within each panel. The yellow and grey maps represent two ab-initio reconstructions, with the yellow map exhibiting severe distortion from one perspective, rendering it unsuitable for use. The squared map chosen for further analysis.

Moreover, we have increased the image size, improved clarity, and adjusted the viewing angles to enhance reader comprehension of our computational process.

We trust these modifications align with your expectations and appreciate your meticulous review.

“An unsharpened map reveals subtle outlines of the receptor-binding domain (RBD) and LamB, and features remain identifiable despite the absence of sharpening.” Where?

Thank you for your attention to detail. The unsharpened map you inquired about is here.

This map is now better positioned and enlarged in the revised Figure S1. We hope this adjustment enhances the visibility and clarity of the subtle outlines of the receptor-binding domain (RBD) and LamB in the unsharpened map.

Figure S3: E: It is not clear what the 5.8Å distance refers to. Reading the legend, I expected it to be from gpl to gpJ in the closed and open states, but the second arrow rather shows gpl closed to open...

The different domains of gpl are not clear.

Thank you for your attention to Figure S3E. The 5.8Å distance and arrow represent the positional changes between a randomly selected atom in gpl and HDs, respectively. This comparison illustrates the similar movement of gpl and nearby HDs in the closed and open states when aligned with gpM. To improve clarity, we have added labels for different domains of gpl in the revised Figure S3 (now is S5).

e

Material and Methods:

What is the final LamB/MSP/lipid ration in mol?

The final LamB/MSP/lipid ratio provided is 1:2.5:290.

The authors thank this referee for his/her time and constructive comments.

Referee #2:

In this manuscript, Ge and Wang present EM structures of the tip of the bacteriophage lambda tail before and after binding to the lambda receptor LamB. This work is of high importance for phage biology and virus research in general (lambda is a very important model virus). The results are of high technical quality. They describe the structures in detail and provide important information about the necessary conformational changes, finishing with a schematic model for infection. The results are very interesting, but I would like to suggest some improvements in the presentation.

1. The work by Breyton et al on another siphovirus (T5) is mentioned and partially referenced, but a detailed comparison is missing. Their important Linares et al, Sci Adv paper is also not cited. I think a detailed discussion of the differences and similarities in the two works is very important.

Thank you for highlighting the importance of comparing our work with the study by Breyton et al. on phage T5. We have thoroughly revised the discussion section to include a detailed comparison of the structures of lambda phage and phage T5. The main differences lie in the structural variances of AHS, CSF (BSP), and RBD, leading to distinctions in the amplification mechanism of their signals upon receptor binding. Your guidance has significantly enriched the comparative analysis in our manuscript.

2. In figure 6 and some earlier figures, it would be useful to include which parts of which steps have now exactly been structurally resolved, i.e. mention the PDB codes in the legends and the new resolved structures boxed.

Thank you for your valuable suggestion. We have updated the figure legends to include information about the newly resolved structures, specifying the relevant PDB codes where applicable.

3. Supplementary movie file(s) of the proposed conformational changes would be very helpful.

Thank you for your suggestion. We have now included supplementary movie files depicting the conformational changes in the AHS, CSF (BSP), and RBD regions. These additions aim to provide a more dynamic and comprehensive visualization of the structural transitions in these key areas.

We appreciate your feedback, and we hope these supplementary materials enhance the understanding of our proposed conformational changes.

4. The description of the structures and conformation changes is a bit dense - I know that it is difficult to write these explanations in a way that is easy to follow, but some improvements could probably be made.

Thank you for your feedback on the density of the structural descriptions. We have revisited both the text and figures, making adjustments to improve clarity and simplify the explanations. Our aim is to facilitate a better understanding of the structures and conformational changes for the readers. We appreciate your input, and we hope the revised presentation meets your expectations.

5. 8XCK-gpJ713 seems to have peptidylprolyl isomerase co-purified and bound in the structure (as identified by BLAST analysis of the sequence) - however I could not find any comment about that in the manuscript or supplementary material. The map also has density at the N-termini that was not modelled - could this be done?

Thank you for bringing this to our attention. The closed gpJ713 structure inadvertently co-purified with the peptidylprolyl isomerase protein during purification. However, upon further analysis, we found that the conformation of the tail tip in the higher-resolution map remained consistent with the low-resolution map obtained from whole tail tip previously (EMDB ID: EMD-35826). Additionally, docking the gpJ713 model into the original low-resolution EM map of whole tail tip revealed that the peptidylprolyl isomerase did not affect the conformation of gpJ713. As the significance of the peptidylprolyl isomerase protein was not understood, it was not discussed further. We have updated the corresponding legends to mention this situation.

6. Fig 3A shows a bigger structure than provided in PDB file 8XCK, where is this structure based on? Mention reference and PDB code. Fig 3C is unclear to me.

Thank you for your inquiry regarding Figure 3A. The model in this figure was constructed using a composite map generated from the complete tail tip complex and the gpJ-713 map. The uploaded structure (PDB code: 8XCK) only includes the high-resolution portion corresponding to the gpJ-713 map. We appreciate your attention to detail and have provided the necessary clarification in the figure caption.

Concerning Figure 3C, we have retained this panel to offer a low-pass surface representation, aiming to assist readers less familiar with phage tail structures in understanding the rotation in the CSF region and the changes in the AHS region.

Additionally, we have included a corresponding movie to further enhance readers' comprehension of this structural aspect. If you have any further suggestions or specific points you'd like us to address, please feel free to let us know.

Minor points:

43: absent should be absence

Thank you, the necessary correction has been made.

111: not sure what is meant by "demonstrates a lateral descent", I suggest rephrasing.

Thank you for your suggestion. The phrase has been revised to "demonstrates a descent."

115: The receptor binding domain...

Thank you, it has been corrected.

Sonnei should not be capitalized, should be sonnei. In two places it is written Sonnei.

Thank you, these necessary corrections have been made.

482: I would include "nanodisc" somewhere in the title of this section

Thank you for your suggestion. The title of the section has been revised to "LamB in nanodisc."

ref 2: title does not need to be capitalised throughout

Thank you, it has been corrected.

Fig 3: outter should be outer

Thank you, it has been corrected.

The authors thank this referee for his/her time and constructive comments.

REVIEWER COMMENTS

Reviewer #1 (Remarks to the Author):

The paper of Ge and Wang improved much upon revision. I guess in my previous review I didn't emphasise that the data are beautiful. I appreciate that the mechanism described is very complicated to explain and that the authors already did a great job explaining and describing it. There are still a few points that would deserve attention to make the paper even clearer:

- In general, I would recommend not to put valuable and unique information in the figure legends. As mentioned, the mechanism is very complicated to explain and understand, and all the information should appear in the main text. Ex: legend to Fig. 3a should be transferred eg. In the paragraph that starts line 153.

- I must have missed it, but I did not see anywhere, in the new version of the text, a rapid description of how the lambda tail tube is closed. Making that clear is really important so that readers can understand how the tail tube opens. This could be included in the last paragraph of the introduction, with reference to figs. S3 (maybe add a longitudinal cross-section view of the closed tube in S3?) and S5c.

- For example, lines 143-147: "The central tail fiber of the lambda phage is constituted by FNIIIs, AHS, CSF, and RBD (Fig. 3a, left). AHS is a helix bundle formed by three α -helices, situated below the FNIIIs, securely anchoring the three FNIII-2 units together from the bottom. The diameter of the tube gradually narrows in the region corresponding to the FNIII, eventually closing completely." As it is a new section, you could recall from which protein you are talking of. Also, you talk of the diameter of the tube: is it the tail tube? I understood it was closed by a combination of gpI plug and HDII-ins2 (as shown in fig. S5c)... Here however, you mention that it closes completely in the FNIII region? Thus, a clear description of how lambda tail tube is closed seems really necessary.

- I really think the paper would benefit from showing a low resolution map of the tip in interaction with the nanodisc, but also with a discussion on the fact that as LamB is a trimer, its cross-section is large, thus leaving less space for a lipid bilayer in the nanodisc, probably explaining why there is no channel visualised across the nanodisc, unlike what is seen with phage T5.

- Fig. 5a,b are still quite difficult to understand. Again, I appreciate the complexity of the mechanism and the effort of the authors to explain it. However, in panel a, gpI seems to be a dimer? Or is it a trimer and the surface representation of a monomer of gpJ hides the third gpI monomer? However, the three-fold symmetry is not visible...

- The supposed intermediate state in panel b might not necessary, as it confuses the reader. The open conformation of gpI should rather be depicted in a new colour, with domain limitation with residue numbers clearly indicated to allow the reader to connect.

- Panels c and d are great! They could be further improved by having gpI have the same colour code as in the other panels. As for panels a and b, in panel e, the three-fold symmetry of gpI is not visible, I am completely confused.

- Fig. S5: here also, the figure is still a bit confusing. In the colour legend, please indicate which protein you are talking about, not only which domain (eg. closed and opened states $\beta 1$ in panel d => gpJ $\beta 1$, as $\beta 1$ appears also in gpI...). In panel e, please do not use the same colour for gpI closed and open state. It is the difficult part to understand, you should help the reader to see very well the initial and final conformation of the protein.

- Lines 255-260: "Following the expulsion of gpH from the tail tube, viral DNA flows through the tube toward the inner membrane and finally enters the host cell for replication²⁸ (Fig. 6, step 4). Despite the insights gained, our study presents some limitations. Further experimental validation is required to confirm this mechanism. Further molecular and structural studies are needed to elucidate these interactions and their implications for the phage life cycle." This conclusion seems a bit awkward. The mechanism described is beautiful and relies on very strong experimental data. Thus, be more precise on what you believe would need to be confirmed and elucidated. For example, you should maybe mention that what remains to be determined is how the peptidoglycan and the inner membrane are perforated.

Minor points:

- Lines 98-99 "Following extensive cryo-EM image processing, both the tail tube and the density outside of the tube were successfully obtained" is not very clear. What is density outside of the tube?
- Lines 147-149: "In the closed state, the longitudinally oriented CSF, a mixed β -sheet prism, exhibits intricate torsion in the strands spanning from the RBD to the CSF." Not clear. If it is the CSF, how does it go from RBD to CSF?
- Lines 206-210 : "The gpI protein consists of several components: an invisible domain on the N-terminal, two helices (H1 and H2), a 'plug' domain (159-175) within the tube lumen, and a series of β -strands interconnected by loops (Figs. 5a, 5b). In the closed state, gpI residues from position 101 onward are visible, but in the open structure, only residues from position 167 onward can be seen (Fig. 5b)." Prefer "domains" to "components", "unresolved domain" to "invisible domain", "resolved" to "visible"?
- Line 226: "Bacteriophage lambda is an ideal model system for studying host recognition and infection trigger mechanisms." This statement without further explanation is a bit weird... All phages are ideal models...
- Line 245: choose whether to write in the past or present tense, not a mixture of both.
- Legend to Fig. 6: "The N-terminal parts of gpI are anchoring to the outer membrane." This has not been proven, please be more cautious.
- Figure 3: note gpJ domain limits in panel b too. Attention, in the inset of panel a-right, do not include LamB, which is not shown in the structures of the panel.
- Figure S4: indicate the LamB PDB codes used for the figure.

Reviewer #2 (Remarks to the Author):

The authors have improved the manuscript and responded to all the points raised. In my opinion, the subject is important and the reported results very significant. I think there can still be some improvements made in the writing, but this can now be handled by editorial staff in concert with the authors.

Response to Referees' Comments

Referee #1:

The paper of Ge and Wang improved much upon revision. I guess in my previous review I didn't emphasise that the data are beautiful. I appreciate that the mechanism described is very complicated to explain and that the authors already did a great job explaining and describing it. There are still a few points that would deserve attention to make the paper even clearer:

We sincerely appreciate your thoughtful review of our manuscript. Your recognition of the improvements made in our paper upon revision is truly encouraging. We are also grateful for your acknowledgment of the complexity of the mechanism described and your appreciation of our efforts in explaining and describing it. It is heartening to hear that you find the data beautiful. We have taken note of your suggestions for further clarity and will ensure that the necessary adjustments are made to enhance the manuscript's readability.

- In general, I would recommend not to put valuable and unique information in the figure legends. As mentioned, the mechanism is very complicated to explain and understand, and all the information should appear in the main text. Ex: legend to Fig. 3a should be transferred eg. In the paragraph that starts line 153.

We appreciate your suggestion regarding the placement of unique information in figure legends. Recognizing the complexity of the mechanism described, we will transfer essential information from figure legends to the main text, as you recommended. Specifically, we will ensure that the content from the legend of Fig. 3a is integrated into the corresponding section in the main text (line 167-171).

- I must have missed it, but I did not see anywhere, in the new version of the text, a rapid description of how the lambda tail tube is closed. Making that clear is really important so that readers can understand how the tail tube opens. This could be included in the last paragraph of the introduction, with reference to figs. S3 (maybe add a longitudinal cross-section view of the closed tube in S3?) and S5c.

Thank you for your comment. We acknowledge the importance of providing a clear description of how the lambda tail tube is closed to facilitate readers' understanding of the tail tube opening mechanism. In the section titled "Tail tip baseplate hub domains (BHD) opening mechanism", located around line 195, it is mentioned that the diameter of the tail tube gradually decreases starting from HDIII-HDII until it is finally sealed by the combined action of FNIII, such as the trimeric FNIII-1 and FNIII-2. Additionally, we now reference Figs. S3 and S5c to provide further clarity, as you suggested. To assist readers, we have included a longitudinal cross-section view of the closed lambda phage tail tip in Figure S3b.

- For example, lines 143-147: "The central tail fiber of the lambda phage is constituted

by FNIII_s, AHS, CSF, and RBD (Fig. 3a, left). AHS is a helix bundle formed by three α -helices, situated below the FNIII_s, securely anchoring the three FNIII-2 units together from the bottom. The diameter of the tube gradually narrows in the region corresponding to the FNIII, eventually closing completely.” As it is a new section, you could recall from which protein you are talking of.

Thank you for your feedback. The central tail fiber region of the lambda phage is entirely composed of the C-terminal portion of the gpJ protein. We have taken note of your suggestion and have made the necessary revisions to the manuscript as per your request. In particular, we have revised lines 143-147 to specify the protein being discussed, ensuring clarity and coherence in the text. This adjustment aligns with your recommendation and improves the overall quality of the manuscript. We appreciate your attention to detail and your dedication to enhancing the clarity of our work.

Also, you talk of the diameter of the tube: is it the tail tube? I understood it was closed by a combination of gpI plug and HDII-ins2 (as shown in fig. S5c)··· Here however, you mention that it closes completely in the FNIII region? Thus, a clear description of how lambda tail tube is closed seems really necessary.

Thank you for your insightful comments. We have carefully considered your concerns regarding the closure of the lambda tail tube and have made appropriate revisions to provide a clearer description in the manuscript. In the revised version, we have elaborated on the closure mechanism of the lambda tail tube.

When in the closed state, the diameter progressively narrows to 20 Å from the HDIII-HDII region until it is eventually sealed through the combined action of FNIII_s. Inside the baseplate cone, the trimeric plug domains of gpI extend into the cavity, effectively obstructing the tube. Upon LamB binding and subsequent dissociation of FNIII_s, the distal end of the tail tip opens. However, the narrowest part of the channel remains restricted to a diameter of only 10 Å (Fig. S5c), which is defined by the HDII-ins-1 domains. Following this, the dissociation of FNIII_s triggers the destabilization of the trimer interaction network of HDII-HDIII, leading to its rigid-body rotational movement away from the tail tip axis. The insertion domains HDII-ins1 and HDII-ins2 follow the rotation of HDII-HDIII, resulting in the opening of the tail tip with a channel approximately 30 Å in diameter from the HDI-HDIV ring to the HDII-ins1 tip. Without support from the baseplate hub domains, the gpI plugs collapse towards the outer membrane of the host cell.

We believe these revisions address your concerns and provide a clearer understanding of the closure mechanism of the lambda tail tube. We appreciate your feedback and your dedication to improving the quality of our manuscript.

- I really think the paper would benefit from showing a low resolution map of the tip in interaction with the nanodisc, but also with a discussion on the fact that as LamB is a trimer, its cross-section is large, thus leaving less space for a lipid bilayer in the nanodisc,

probably explaining why there is no channel visualised across the nanodisc, unlike what is seen with phage T5.

Thank you for your suggestion. We have addressed this by including a low-resolution map of the tail tip in interaction with LamB in a nanodisc in Figure S1a. We have also referenced your insight in the manuscript, stating: "After further classification of the particles ultimately used, we also obtained a low-resolution map depicting the interaction between the tail tip and LamB within a nanodisc. However, the majority of particles did not fall into this category. This may be attributed to the nanodisc scaffold MSP2N2 being insufficient to simultaneously encapsulate the LamB trimer and the membrane-inserted tube end." We appreciate your valuable input and believe these additions enhance the clarity and completeness of our manuscript.

- Fig. 5a,b are still quite difficult to understand. Again, I appreciate the complexity of the mechanism and the effort of the authors to explain it. However, in panel a, gpI seems to be a dimer? Or is it a trimer and the surface representation of a monomer of gpJ hides the third gpI monomer? However, the three-fold symmetry is not visible...

Thank you for your feedback regarding Fig. 5a and b. We acknowledge the complexity of the mechanism depicted in these figures and appreciate your efforts to understand and provide constructive criticism.

In response to your query, we have carefully reviewed and revised Fig. 5a and b to improve clarity. Specifically, we have adjusted the perspective and increased the transparency of the surface representation of gpJ monomer, while also providing clarification in the corresponding figure caption that the depiction shows a gpI trimer cartoon and the surface of one gpJ HDs. Additionally, in Fig. 5e, we have opted not to display the surface of gpJ to emphasize that gpI is a trimer.

- The supposed intermediate state in panel b might not necessary, as it confuses the reader. The open conformation of gpI should rather be depicted in a new colour, with domain limitation with residue numbers clearly indicated to allow the reader to connect.

In response to your comment regarding Fig. 5b, we have removed the intermediate state and utilized different colors for the open conformation and closed gpI to distinguish between them. Additionally, we have included clear domain limitations with residue numbers to facilitate better understanding and connection for the reader.

- Panels c and d are great! They could be further improved by having gpI have the same colour code as in the other panels. As for panels a and b, in panel e, the three-fold symmetry of gpI is not visible, I am completely confused.

In response to your comment, we have revised panels c and d to ensure that gpl has the same color code as in the other panels, thus maintaining consistency throughout the figure.

We understand your concern regarding the visibility of the three-fold symmetry of gpl in panel e. We have carefully reviewed these panels and made adjustments to enhance clarity. Specifically, we have ensured that gpl in panel e accurately reflects the three-fold symmetry to avoid confusion.

We believe that these revisions will significantly enhance the comprehensibility of Fig. 5a, b, c, d and e, facilitating a better understanding of the mechanism presented. We are grateful for your valuable input and your dedication to improving the quality of our manuscript.

- Fig. S5: here also, the figure is still a bit confusing. In the colour legend, please indicate which protein you are talking about, not only which domain (eg. closed and opened states $\beta 1$ in panel d => gpJ $\beta 1$, as $\beta 1$ appears also in gpl...). In panel e, please do not use the same colour for gpl closed and open state. It is the difficult part to understand, you should help the reader to see very well the initial and final conformation of the protein.

Thank you for your feedback regarding Fig. S5. We acknowledge your concerns regarding the clarity of the figure and appreciate your suggestions for improvement.

In response to your comment, we have made the following revisions:

In the color legend, we have indicated which protein is being referred to, not only which domain. For example, in panel d, we have specified "closed and opened states $\beta 1$ " as "gpJ $\beta 1$," as $\beta 1$ also appears in gpl.

In panel e, we have ensured that different colors are used for gpl closed and open states to distinguish them more clearly. This adjustment aims to facilitate a better understanding of the initial and final conformation of the protein.

These revisions are intended to address your concerns and improve the clarity of Fig. S5. We appreciate your valuable feedback and your dedication to enhancing the quality of our manuscript.

- Lines 255-260: "Following the expulsion of gpH from the tail tube, viral DNA flows through the tube toward the inner membrane and finally enters the host cell for replication²⁸ (Fig. 6, step 4). Despite the insights gained, our study presents some limitations. Further experimental validation is required to confirm this mechanism. Further molecular and structural studies are needed to elucidate these interactions and their implications for the phage life cycle." This conclusion seems a bit awkward. The mechanism described is beautiful and relies on very strong experimental data. Thus, be more precise on what you believe would need to be confirmed and elucidated. For

example, you should maybe mention that what remains to be determined is how the peptidoglycan and the inner membrane are perforated.

Thank you for your suggestion. The sentence has been changed as “Despite the insights gained, our study presents some limitations. The enzyme from the lambda phage that is responsible for cleaving the cell wall peptidoglycan in the periplasmic space remains unidentified, though it is speculated to possibly be the N-terminal part of gpI. Additionally, the mechanism by which lambda DNA traverses the periplasmic space and inner membrane is still unknown. Further molecular and structural studies are needed to elucidate these interactions and their implications for the phage life cycle.”

Minor points:

- Lines 98-99 “Following extensive cryo-EM image processing, both the tail tube and the density outside of the tube were successfully obtained” is not very clear. What is density outside of the tube?

Thank you for your feedback regarding lines 98-99. We have revised the statement to address the clarity issue. The revised version now reads: "Following extensive cryo-EM image processing, both the tail tube and the density of the FNIII of gpJ were successfully obtained." This revision provides a clearer description of the obtained data and avoids ambiguity. We appreciate your attention to detail and your commitment to improving the clarity of our manuscript.

- Lines 147-149: “In the closed state, the longitudinally oriented CSF, a mixed β -sheet prism, exhibits intricate torsion in the strands spanning from the RBD to the CSF.” Not clear. If it is the CSF, how does it go from RBD to CSF?

Thank you for your suggestion. The revised statement now reads: "In the closed state, the longitudinally oriented CSF, a mixed β -sheet prism, exhibits intricate torsion in the strands spanning from above the RBD to below the AHS."

- Lines 206-210 : “The gpI protein consists of several components: an invisible domain on the N-terminal, two helices (H1 and H2), a 'plug' domain (159-175) within the tube lumen, and a series of β -strands interconnected by loops (Figs. 5a, 5b). In the closed state, gpI residues from position 101 onward are visible, but in the open structure, only residues from position 167 onward can be seen (Fig. 5b).” Prefer “domains” to “components” , “unresolved domain” to “invisible domain” , “resolved” to “visible” ?

Thank you for your feedback regarding lines 206-210. We have incorporated your suggestions into the revised version as follows: “The gpI protein consists of several domains: an unresolved domain on the N-terminal, two helices (H1 and H2), a 'plug' domain (159-175) within the tube lumen, and a series of β -strands interconnected by loops (Figs. 5a, 5b). In the closed state, gpI residues from position 101 onward are

resolved, but in the open structure, only residues from position 167 onward can be seen (Fig. 5b).”

- Line 226: “Bacteriophage lambda is an ideal model system for studying host recognition and infection trigger mechanisms.” This statement without further explanation is a bit weird... All phages are ideal models...

Thank you for your feedback regarding the statement about Bacteriophage lambda. We have revised the statement to address your concern. The revised version now reads: "Bacteriophage lambda has been widely utilized as a model system for studying host recognition and infection trigger mechanisms." We appreciate your attention to detail and your commitment to improving the clarity of our manuscript.

- Line 245: choose whether to write in the past or present tense, not a mixture of both.

Thank you for your observation regarding the tense in line 245. We have revised the text to ensure consistency, now using the present tense throughout.

- Legend to Fig. 6: “The N-terminal parts of gpI are anchoring to the outer membrane.” This has not been proven, please be more cautious.

Thank you for bringing this to our attention. We have removed the statement from the legend to Fig. 6 to ensure accuracy and avoid any unsupported claims.

- Figure 3: note gpJ domain limits in panel b too. Attention, in the inset of panel a-right, do not include LamB, which is not shown in the structures of the panel.

Thank you for your valuable feedback on Figure 3. In panel b, we have included gpJ domain limits to enhance clarity and aid comprehension. In the inset of panel a-right, we have removed LamB from the black box in the schematic diagram. These changes have been implemented to improve the accuracy and coherence of Figure 3. Thank you for your attention to detail.

- Figure S4: indicate the LamB PDB codes used for the figure.

Thank you for your suggestion regarding Figure S4. We have now included the LamB PDB codes as requested. Your feedback is greatly appreciated.

The authors thank this referee for his/her time and constructive comments.

Referee #2:

The authors have improved the manuscript and responded to all the points raised. In my opinion, the subject is important and the reported results very significant. I think there can still be some improvements made in the writing, but this can now be handled by editorial staff in concert with the authors.

Thank you for your positive feedback and recognition of the enhancements made to the manuscript. We are delighted that you consider the subject matter important and the results significant. Your suggestion for further writing improvements is appreciated, and in response to feedback from reviewer #1, we have refined the text accordingly. Ensuring clarity and coherence in presenting our research findings remains our priority. We sincerely appreciate your valuable feedback and support throughout the review process.

The authors thank this referee for his/her time and constructive comments.

REVIEWERS' COMMENTS

Reviewer #1 (Remarks to the Author):

I thank the authors to have clarified the figures and the text, so that the complex mechanism of tail tube opening now appears much more comprehensible.

Minor points (that need to be corrected before publication):

- Fig. S3: I guess there is a miss-label of gpJ-HDIV in panel b as it has the colour code of gpL.
- "- Figure 3: note gpJ domain limits in panel b too. Attention, in the inset of panel a-right, do not include LamB, which is not shown in the structures of the panel.

Thank you for your valuable feedback on Figure 3. In panel b, we have included gpJ domain limits to enhance clarity and aid comprehension."

I think this was skipped as no domain limits can be seen in panel 3b.

- The paragraphs that goes from line 161 to line 180 needs to be rewritten a little to accommodate and integrate the copy-paste text from the figure legend (shortening of what?) and make a coherent description of the mechanism. Also, in this paragraph and in the rest of the document, please, do not re-define abbreviations already defined earlier.

Response to Referees' Comments

Referee #1:

I thank the authors to have clarified the figures and the text, so that the complex mechanism of tail tube opening now appears much more comprehensible.

Thank you for your thoughtful review and valuable feedback on our manuscript. We appreciate your acknowledgment of the improvements made to clarify both the figures and the text.

Minor points (that need to be corrected before publication):

- Fig. S3: I guess there is a miss-label of gpJ-HDIV in panel b as it has the colour code of gpL.

We acknowledge the mislabeling of the gpJ-HDIV in panel b of Fig. S3 and have corrected it accordingly. Thank you for bringing this to our attention.

- "- Figure 3: note gpJ domain limits in panel b too. Attention, in the inset of panel a-right, do not include LamB, which is not shown in the structures of the panel.

Thank you for your valuable feedback on Figure 3. In panel b, we have included gpJ domain limits to enhance clarity and aid comprehension."

I think this was skipped as no domain limits can be seen in panel 3b.

We have now updated Figure 3b to incorporate these limits for enhanced clarity.

- The paragraphs that goes from line 161 to line 180 needs to be rewritten a little to accommodate and integrate the copy-paste text from the figure legend (shortening of what?) and make a coherent description of the mechanism. Also, in this paragraph and in the rest of the document, please, do not re-define abbreviations already defined earlier.

We have carefully revised the paragraphs from line 161 to line 180 to integrate the copy-pasted text from the figure legend and ensure a coherent description of the mechanism. We have also refrained from redefining abbreviations already defined earlier in the document, as per your guidance.

The authors thank this referee for his/her time and constructive comments.